# Seeing What's Wrong: A Trajectory-Guided Approach to Caption Error Detection

**Gabriel Afriat**[1], **Ryan Lucas**[1], **Xiang Meng**[1], **Yufang Hou**[2,3], **Yada Zhu**[3], **Rahul Mazumder**[1]

[1]Massachusetts Institute of Technology, [2]IT:U Interdisciplinary Transformation University Austria, [3]IBM Research

{afriatg,ryanlu,mengx,rahulmaz}@mit.edu, yufang.hou@it-u.at, yzhu@us.ibm.com

## Abstract

Error detection is critical for enhancing multimodal dataset reliability and downstream model performance. Existing error filters, while increasingly powerful, typically rely on a single similarity score per image–caption pair. This is limiting: captions with subtle errors (e.g., mislabeled objects, incorrect colors, or negations) can still score highly, while correct but imprecisely worded captions may score poorly. To address this, we introduce the notion of a *caption trajectory*: an ordered sequence of captions produced by iteratively editing a caption to maximize an image-text relevance score. This trajectory carries rich signals for error detection. Correct captions typically stabilize after minor edits, while erroneous captions undergo substantial improvements. Building on these insights, we introduce *TRACED*, a cost-efficient and model-agnostic framework that leverages trajectory statistics for more accurate caption error detection. Beyond detection, *TRACED* also serves as an interpretable tool for identifying the origins of errors. We further demonstrate that, in the case of error correction, this interpretable token-level error information can be provided to VLMs to enhance the alignment score of the generated captions. On MS COCO and Flickr30k, *TRACED* achieves up to 2.8% improvement in accuracy for error detection across three noise types. Our code is available at https://github.com/mazumder-lab/TRACED.

## 1 Introduction

Vision models have achieved remarkable success across diverse applications, including visual understanding (Dosovitskiy et al., 2021), multimodal reasoning (Alayrac et al., 2022), and content generation (Esser et al., 2024). These models require extensive training on massive datasets, often containing millions of image-caption pairs (Ordonez et al., 2011; Lin et al., 2014; Russakovsky et al., 2015; Schuhmann et al., 2021; Bain et al., 2021; Changpinyo et al., 2021; Li et al., 2022). However, many rely on pre-training with web-scraped (Radford et al., 2021; Li et al., 2021; Lin et al., 2024) or even synthetic data (Li et al., 2022; 2023; Hammoud et al., 2024). These datasets often contain significant errors (Northcutt et al., 2021b; Liao et al., 2021; Zhang et al., 2025), which not only hampers model convergence during training but can also reinforce biases and reduce generalization capabilities. Recent studies have demonstrated that removing incorrect image-caption pairs (Zhang et al., 2025; Li et al., 2022) can substantially improve model performance. Therefore, detecting such errors is essential for boosting data quality and training better models.

As manual annotation is infeasible at scale, many works have proposed automated error detection. Existing error detection methods typically rely on assigning a quality or similarity score to each image-caption pair, using either model confidence (Pleiss et al., 2020; Swayamdipta et al., 2020; Northcutt et al., 2021a), neighborhood consistency (Bahri et al., 2020; Zhu et al., 2022; Zhang et al., 2025), or multimodal alignment (Radford et al., 2021; Li et al., 2022; Zhang et al., 2025). While these existing methods are increasingly powerful, they typically rely on a single similarity score per image-caption pair. This poses a key limitation: *not all errors are equally detectable*. Some captions may mostly align with the image but include subtle mistakes—incorrect object labels, color description, or negation—that still yield high similarity scores. Conversely, a correct caption might receive a low score if the image is difficult to describe or if the wording is imprecise (see Figure 1). In both cases, relying on a single similarity score can lead to unreliable error detection.

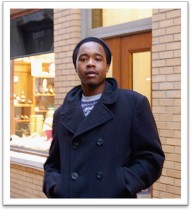

Noisy Caption: A man is standing in front of a brick storefront wearing no jacket.

BLIP Alignment Score: 0.55 **(no error)**

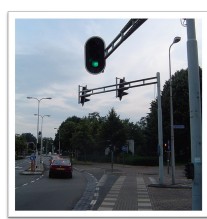

Correct Caption: Vehicles on a street near a green traffic light.

BLIP Alignment Score: 0.44 **(error)**

Figure 1: Left: the BLIP-based alignment score (ITM block) is high (above 0.5), likely because the caption is mostly correct except for a single erroneous word ("no"). Right: the BLIP-based alignment score is low (below 0.5) even though the caption is correct.

In this paper, we propose a novel approach that leverages caption improvement trajectories for more accurate error detection. Our key insight is that *the potential for improvement varies significantly between correct and incorrect captions*, a pattern we observe consistently across the state-of-the-art alignment scoring functions we evaluated. Specifically, when starting with an accurate caption, iterative attempts to improve it yield minimal gains in similarity scores. In contrast, an incorrect caption presents substantial improvement potential. We formalize these intuitions by generating a sequence of increasingly refined captions for each image-caption pair and analyzing the resulting trajectory. Rather than making error detection decisions based on a single similarity score, our method examines the *pattern of improvement across the entire sequence*. Importantly, this trajectory-based framework is model-agnostic and can be combined with existing state-of-the-art error detection baselines to enhance their performance.

To evaluate error detection in image captioning, synthetic noise is typically injected through caption swaps (Zhang et al., 2025). More sophisticated swaps (e.g., between captions sharing nouns or metadata) can create harder cases but still often yield captions that are unrelated to the image and easy to detect. To test our framework under more challenging and realistic conditions, we introduce a new form of fine-grained noise generated with GPT-4o-mini (OpenAI, 2024). By prompting the model to make minimal yet semantically significant alterations, this noise yields captions that remain plausible but contain subtle errors, making them harder to detect (see left example in Figure 1).

We further show that our error detection framework has broader applicability. In particular, it provides interpretable insights into the origin of errors, which can then be leveraged to guide a VLM toward potential error sources, thereby correcting them and enhancing the alignment score of the edited captions.

Our contributions are as follows:

1. We introduce a new error detection framework called *TRACED* (**Tra**jectory **C**reation for **E**rror **D**etection), based on the novel idea of creating caption trajectories. By iteratively improving captions through token replacements and deletions, we generate a sequence of captions and analyze both their alignment with the corresponding image and the semantic changes between iterations. This trajectory-based approach provides richer signals and enables more accurate identification of mismatched image-caption pairs. *TRACED* is cost-efficient and interpretable. It is also flexible and can be applied on top of many existing error detection methods to enhance their performance.

2. We evaluate how *TRACED* improves the performance of several state-of-the-art error detection methods, including CLIP (Radford et al., 2021), LEMON (Zhang et al., 2025), and BLIP (Li et al., 2022). Our experiments contain various types of label noise, including traditional random caption swaps and a more challenging type of synthetic noise we generated by prompting GPT-4o-mini (OpenAI, 2024). Compared to the existing benchmarks, this novel type of noise consists of plausible yet incorrect captions designed to better reflect real-world annotation errors. On average across all noise types, *TRACED* consistently improves detection Accuracy by up to 2.5% on MS COCO (Lin et al., 2014), 2.8% on Flickr30k (Plummer et al., 2015), and 2.4% on MM-IMDb (Arevalo et al., 2017).

3. We show how *TRACED* can be used to provide interpretable outputs and identify specific misaligned tokens in erroneous captions. On InternVL3 models, we evaluate the impact

of this interpretable token-level error information on caption correction. We show that this information can be used to improve the alignment of the generated captions, and observe an improvement of up to 14.5% in the BLIP-alignment score for the corrected captions using *TRACED* compared to unguided caption correction.

## 2 RELATED WORK

**Handling Noise in Vision Datasets.** Vision datasets often contain labeling errors that degrade model performance (Zhang et al., 2021; Northcutt et al., 2021b; Liao et al., 2021; Zhang et al., 2025). To address this problem, two main research directions have emerged: (i) learning with noisy labels by adapting the loss function or reducing the influence of likely corrupted pairs (Natarajan et al., 2013; Bennouna et al., 2023; Arazo et al., 2019; Huang et al., 2023), and (ii) data cleaning, which aims to detect and remove mislabeled samples (Grivas et al., 2020; Zhang et al., 2025). Our work follows the second line, improving the filtering of noisy image–caption pairs.

**Error Detection for Classification Datasets.** Label noise can be detected through various approaches. Confidence-based methods such as Confident Learning (Northcutt et al., 2021a), AUM (Pleiss et al., 2020), and Dataset Cartography (Swayamdipta et al., 2020) flag mislabeled samples based on model confidence. Neighbor-based approaches, including Deep k-NN (Bahri et al., 2020) and SimiFeat (Zhu et al., 2022), detect label noise by checking agreement with nearest neighbors in an embedding space. With the emergence of foundation models, new stronger baselines for label error detection have appeared. Liang et al. (2024) and Kang et al. (2022) propose leveraging CLIP (Radford et al., 2021), pretrained on 400M image-text pairs, to score image-label consistency. Building on this, LEMoN (Zhang et al., 2025) introduces a neighborhood-based method that aggregates relevance scores from multimodal nearest neighbors to improve error detection in classification and image captioning datasets. It outperforms prior confidence- and neighborhood-based methods, making it a strong baseline, which we further enhance with our trajectory-based framework.

**Error Detection for Image Captioning.** In this paper, we focus on error detection in image captioning, a more challenging task than image classification, as it requires a deeper semantic understanding of both language and visual content. To improve caption quality, BLIP (Li et al., 2022) builds on CLIP by learning a shared image-text embedding space but also by training a classifier to distinguish high-quality from noisy image-caption pairs. Although not originally intended for error detection, Zhang et al. (2025) show that BLIP's filtering component performs very well in identifying mislabeled image–caption pairs on the dataset it was fine-tuned on. We therefore also examine how our framework can further enhance BLIP on caption error detection.

**Evaluation via Synthetic Noise Injection.** Error detection methods are often evaluated by injecting synthetic label noise into clean datasets. Prior work has studied symmetric noise, where labels are randomly swapped (Pleiss et al., 2020; Kang et al., 2022), asymmetric noise, where labels are replaced with semantically similar ones via a transition matrix (Northcutt et al., 2021a), and instance-dependent noise, where incorrect labels depend on instance features (Liang et al., 2024; Zhu et al., 2022). These noise models, however, are designed for classification tasks. Zhang et al. (2025) extend noise modeling to image captioning via random caption swaps, swaps between captions with shared nouns, and swaps within the same category using metadata. While more realistic, these approaches still replace entire captions, producing descriptions that can be unrelated to the image. In practice, noise is often subtler: annotators may describe the correct image but misrepresent specific elements, and provide captions that are mostly accurate yet partially wrong. In this paper, we introduce another type of noise for image captioning that aims at capturing this fine-grained form of caption noise and evaluate our framework in this more challenging setting.

**Error Correction for Image Captioning.** Prior work on caption correction has followed two main directions: (a) structured editing frameworks, where models are trained to add or delete words in a sentence (Wang et al., 2022), and (b) dedicated correction models designed to fix a caption (Sammani & Melas-Kyriazi, 2020; Huang et al., 2024; Berger et al., 2025). Our framework has direct applications to the second line of work. Early efforts used LSTMs for correction (Sammani & Melas-Kyriazi, 2020), while later work fine-tuned small VLMs (Berger et al., 2025) or leveraged large closed-source models (Huang et al., 2024) such as GPT-4 (OpenAI et al., 2024) . These studies highlight the effectiveness of transformer-based architectures for caption correction, with large models offering strong but costly performance, and smaller VLMs providing an efficient alterna-

tive for large-scale dataset cleaning. In this paper, we study both settings: we evaluate *TRACED* on InternVL3-14B, the leading open-source VLM under 20B parameters on the OpenVLM Leaderboard (Duan et al., 2024), and on smaller fine-tuned versions of InternVL3-1B.

## 3   *TRACED*: A TRAJECTORY-BASED FRAMEWORK FOR ERROR DETECTION

To address the limitations of single-score image-caption alignment methods, we propose *TRACED*, a trajectory-based framework that leverages iterative caption refinement for improved error detection. *TRACED* iteratively modifies the caption to increase its alignment with the image and tracks how alignment evolves across these edits. This produces a *caption trajectory*, i.e. a sequence of increasingly refined captions, which we use as a signal for error detection. Our core insight is as follows: (i) If the original caption is correct, alignment scores should improve only slightly, and edits will leave the meaning largely intact. (ii) If the caption is incorrect, alignment can typically be improved substantially—often requiring major revisions. By capturing how easily and meaningfully a caption can be improved, *TRACED* provides a richer and more interpretable signal than any single similarity score. *TRACED* is flexible and can be integrated with any existing scoring-based error detection method. *TRACED* can be used for interpretability: using trajectory statistics, we can identify tokens that are likely incorrect in a caption. Another application of *TRACED* is caption correction. The interpretable information on the origin of the error can be provided to VLMs to guide caption correction and improve the alignment score of the generated captions.

### 3.1   TRAJECTORY GENERATION AND ASSESSMENT

**Trajectory Creation and Evaluation.** Let $\mathcal{X}$ denote the set of captions and $\mathcal{Y}$ the set of images. We assume access to a relevance scoring function:

$$s: \quad \mathcal{X} \times \mathcal{Y} \longrightarrow \mathbb{R}$$
$$(x, y) \mapsto s(x, y)$$

This function assigns a real-valued relevance score to an image-caption pair, with higher values indicating stronger alignment. The choice of $s$ is flexible: it may represent the matching probability in BLIP (Li et al., 2022), the cosine similarity of CLIP image and text embeddings (Kang et al., 2022), or a multimodal similarity metric like LEMoN (Zhang et al., 2025).

To capture how a caption evolves during the procedure, we define a trajectory evaluation function:

$$e: \quad \mathcal{X}^{T+1} \times \mathcal{Y} \longrightarrow \mathbb{R}^d$$
$$(x_0, \ldots, x_T, y) \mapsto e(x_0, \ldots, x_T, y)$$

where $T + 1$ is the trajectory length and $d$ is the dimensionality of the trajectory representation used for error detection.

A simple choice of $e$ is the concatenation of relevance scores: $e(x_0, \ldots, x_T, y) = [s(x_0, y), \ldots, s(x_T, y)]$.

Another interesting metric to keep track of is the semantic similarity between the caption at step $t$ and the original (potentially noisy) caption $x_0$, denoted $c(x_t, x_0)$. This captures the degree of semantic change introduced at each step.

In this paper, we focus on these two key signals and construct the following evaluation function:

$$e(x_0, \ldots, x_T, y) = [s(x_0, y), \ldots, s(x_T, y),$$
$$c(x_1, x_0), \ldots, c(x_T, x_0)]$$

While any metrics can be used here, we observe in Appendix A.1 that relying solely on $s$ or $c$ for example is suboptimal. Using both is consistently better. Given access to $s$ and $e$, *TRACED* constructs and evaluates a caption trajectory as described in Algorithm 1.

---

**Algorithm 1** Trajectory Creation and Evaluation

**Input:** initial caption $x$, image $y$, scoring function $s$, evaluation function $e$, number of exploration steps $T$, number of caption candidates $N$ at each exploration step
Initialize $x_0 \leftarrow x$
**for** $t = 1$ **to** $T$ **do**
   Generate candidate alternatives: $x_t^{(1)}, \ldots, x_t^{(N)}$
   Select best candidate:
   $j_t \leftarrow \arg\max_{j \in [N]} s(x_t^{(j)}, y)$
   Set $x_t \leftarrow x_t^{(j_t)}$
**end for**
**Output:** $e(x_0, \ldots, x_T, y)$

---

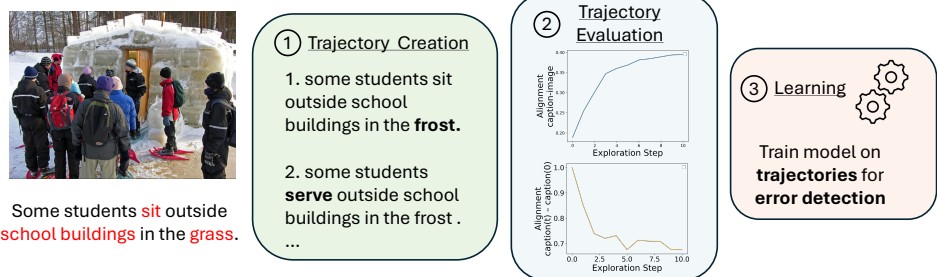

Figure 2: *TRACED* Pipeline for error detection on an example from Flickr30k (Plummer et al., 2015). Given a noisy image-caption pair, a caption trajectory is generated by iteratively maximizing a relevance scoring function $s$. The trajectory is then evaluated using various alignment metrics, which serve as features to distinguish between correct and incorrect image-caption pairs.

**Learning From Trajectories**. We apply Algorithm 1 to each image-caption pair in the dataset. From the resulting trajectory embeddings, we train a classifier to distinguish between correct and erroneous pairs. The overall framework is described in Figure 2.

### 3.2 CAPTION EXPLORATION

A critical component of Algorithm 1 is the generation of candidate captions at each step. We explore and evaluate several strategies for this purpose:

- **Elimination (Elim).** This simple and efficient method generates candidates by removing one token at a time from the current caption. Formally, for a caption $x = (w_1, \ldots, w_L)$ with $L$ tokens, we set $N = L$ in Algorithm 1 and produce $L$ candidates:

$$x^{(i)} = (w_1, \ldots, w_{i-1}, w_{i+1}, \ldots, w_L)$$

  This strategy is computationally cheap: it requires only $L$ forward passes through the scoring function $s$ in Algorithm 1 and no gradient computations.

- **Greedy Coordinate Descent (GCD).** Inspired by Zou et al. (2023), this method aims to find improved captions by replacing individual tokens with alternatives that increase the relevance score $s$. For each token in a caption of length $L$, we consider the top-$K$ gradient-guided replacements, leading to a candidate pool of size $KL$. Since this is often too large to evaluate exhaustively, we randomly sample $N$ token replacements from this space.

- **Fast GCD (FGCD).** To balance the efficiency and quality of the caption trajectory, we introduce a hybrid strategy that combines Elimination with Greedy Coordinate Descent (GCD). We first apply the Elimination method to identify the token whose removal most improves the relevance score. Then, we explore only the top-$K$ replacements for that specific token, reducing the search space to $K$ candidates. This approach requires only one gradient computation and $K + L$ forward passes per iteration in Algorithm 1, a significant reduction compared to the $KL$ evaluations needed for full exploration. Moreover, by focusing on the most impactful token, we promote more effective substitutions than would be achieved by randomly sampling from a large candidate pool.

The full algorithm descriptions are provided in Appendix A.2.

### 3.3 INTERPRETABILITY

Examining the caption trajectory can help identify the source of the error. As shown in Figure 3, the first tokens whose removal or replacement leads to the greatest improvement in alignment score often correspond to the source of the misalignment.

In this example, the initial alignment score from BLIP's classifier is 0.55, indicating a 55% probability that the image-caption pair is correct. Relying on this score would result in misclassifying the image-caption pair. However, the trajectory shows that a meaningful semantic change can

increase the alignment score to around 99.4%, indicating that the original caption was likely erroneous.

On the contrary, for correct captions, the improvements in alignment scores are often associated with minor semantic changes, as seen on Figure 6 in Appendix A.3. Similar observations can be made when using the GCD and Fast GCD algorithms. Wrong captions are often improved through substitutions with semantically different words, and correct captions tend to be refined with minor edits. Full GCD and Fast GCD trajectories are provided in Figures 7 and 8 in Appendix A.3.

### 3.4 APPLICATION TO CAPTION CORRECTION

**Zero-shot Correction.** The trajectories produced by *TRACED* contain rich information, particularly about the origin of errors. While interpretability can also be derived from GCD and FGCD (by examining which tokens are replaced), we adopt the Elimination Algorithm because it is both computationally efficient and achieves performance comparable or even superior to GCD and Fast GCD, suggesting that the extracted signals are of high quality. In this procedure, we remove one token at a time and observe the change in alignment score: increases suggest that the token is likely incorrect (it is better to remove it), while decreases suggest that it is likely correct. This token-level error localization is then provided to the VLM in the

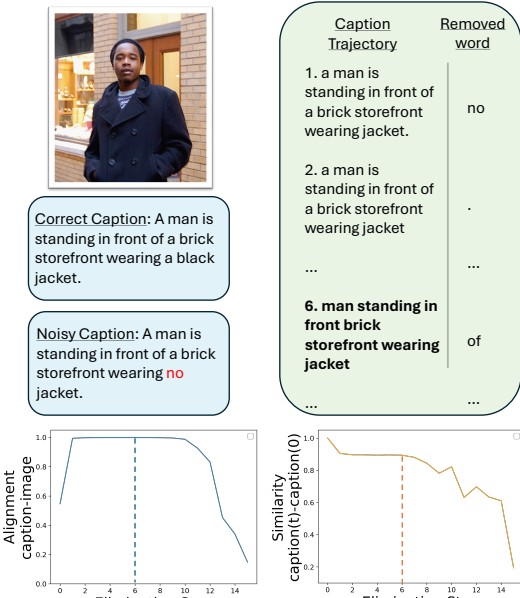

Figure 3: *TRACED* offers interpretability on a Flickr30k example (Plummer et al., 2015), identifying "no" as the source of misalignment. The BLIP-based alignment score (ITM block) peaks at step 6, where the caption accurately matches the image. Removing "no" leads to a notable decline in semantic alignment in the caption trajectory.

prompt, as illustrated in Figure 4. We also investigate the use of Chain-of-Thought (CoT) prompting to test whether reasoning over the flagged words can further aid correction. The exact prompts used are provided in Appendix A.17. In our experiments, we find that both the number of correction prompts and the way the token-level error information is provided play an important role in the final alignment scores. More information on the exact procedure we used is available in Appendix A.5.

**Correction After Fine-tuning.** For large-scale data cleaning, smaller VLMs are advantageous due to their lower cost and faster inference. When sufficient data is available, a small model can be fine-tuned for caption correction using a larger model as teacher. Given the alignment score gains observed for InternVL3-14B when applying *TRACED* in the zero-shot setting (see Figure 5), we

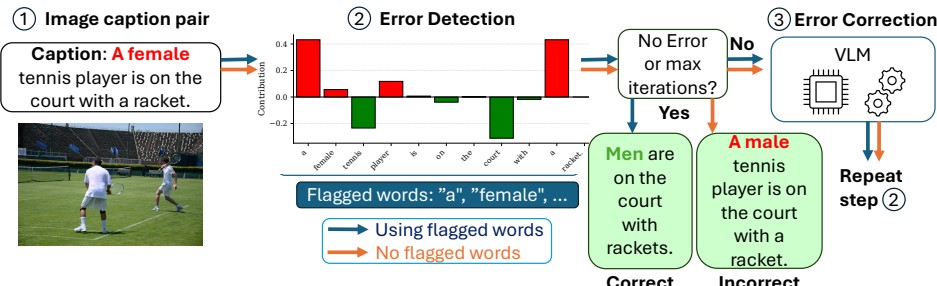

Figure 4: *TRACED* correction pipeline on an example from MS-COCO (Lin et al., 2014). The Elimination procedure identifies words whose removal increases the alignment score (in red on the bar plot). These flagged words are then provided as hints to the VLM, and the process is repeated until no further errors are detected or the maximum number of iterations is reached. Following the hint-guided path (blue) leads to captions with fewer errors compared to the unguided path (orange).

fine-tune InternVL3-1B on captions corrected by InternVL3-14B using *TRACED* 's token-level guidance in the prompts. The resulting model is referred to as InternVL3-1B-FT. At inference time, we isolate the contribution of *TRACED* by comparing two use-cases on the same fine-tuned model: (i) *TRACED*-guided inference, which supplies *TRACED* trajectory information to the VLM, and (ii) a baseline InternVL3-1B-FT model, which uses only the image and noisy caption. This design controls for distillation effects and measures the incremental benefit of *TRACED* at inference. Additional details on the fine-tuning process of InternVL3-1B is available in Appendix A.6.

## 4 EXPERIMENTS

### 4.1 SETUP

All experiments are conducted using 4 NVIDIA L40 GPUs, each with 40GB of memory. *TRACED* is highly parallelizable (see Appendix A.7). Thus, the datasets are split into 4 subsets, with each GPU processing one subset independently. Sentences are processed in batches of size 128 on each GPU. *TRACED* is implemented using PyTorch (Paszke et al., 2019). Details on the computation overhead are provided in Appendix A.8. In general, *TRACED* is efficient and scalable: with BLIP (ITM), the most computationally expensive baseline, *TRACED*-BLIP classifies 1,000,000 image–caption pairs in roughly 6.5 hours on 4 L40 GPUs using the Elimination algorithm.

### 4.2 BASELINES AND DATASETS

We evaluate *TRACED* against common error detection baselines on benchmark datasets by injecting noise into previously clean captions, with the goal of identifying erroneous image–caption pairs.

**Baselines.** We consider BLIP (Li et al., 2022), LEMoN (Zhang et al., 2025) and CLIP (Radford et al., 2021; Kang et al., 2022). CLIP uses cosine similarity in a joint embedding space, LEMoN aggregates CLIP scores from nearest neighbors, and BLIP combines contrastive learning (ITC block) to learn a shared image-text embedding space with a classification head (ITM block) for alignment prediction. LEMoN supports two versions: FIX (default hyperparameters) and OPT (hyperparameters tuned via validation).

**Integration with *TRACED*.** We apply *TRACED* on top of each of these baselines by using their respective alignment scores as the scoring function $s$ during trajectory construction. For BLIP, we evaluate *TRACED* using both the ITC and ITM modules. For LEMoN, we apply our method to both the FIX and OPT variants. For CLIP, we use the standard cosine similarity between image and text embeddings.

**Datasets.** We evaluate the impact of *TRACED* on LEMoN and CLIP using Flickr30k (Plummer et al., 2015), MS COCO (Lin et al., 2014) and MM-IMDb (Arevalo et al., 2017). For Flickr30k and MS COCO, we use the standard Karpathy split (Karpathy & Li, 2015). For MM-IMDb, we adopt the same random 80/10/10 train-validation-test split as Zhang et al. (2025).

For BLIP, finetuned models are publicly available only for Flickr30k and MS COCO. Therefore, we evaluate the improvements from *TRACED* on these two datasets only.

**Noise Types.** We evaluate *TRACED*'s improvements under three types of synthetic label noise, introducing 50% erroneous image-caption pairs for each seed:

- Random noise: A subset of captions is randomly replaced with others from the dataset.
- Noun noise: Captions are swapped with others that share at least one noun, introducing partial semantic overlap.
- Fine-grained noise: Captions are minimally perturbed using `gpt-4o-mini` to introduce subtle semantic inconsistencies, as described in Appendix A.4. Some illustrative examples of the generated errors are provided in Appendix A.16.

Due to the higher cost of generating fine-grained noise using the ChatGPT API, we limit its use to Flickr30k and MS-COCO. For both random and noun noise, we follow the methodology introduced in Zhang et al. (2025).

## 4.3 Experimental details for trajectory construction and learning framework

**Trajectory Generation Hyperparameters.** The trajectory generation hyperparameters for Elimination, GCD, and Fast GCD are detailed in Appendix A.9.

**Trajectory Evaluation Metrics.** For the alignment score $s(x_t, y)$, we use the scoring function of the baseline being evaluated — either CLIP, LEMoN, or BLIP. For the semantic similarity $c(x_t, x_0)$, we compute the cosine similarity between the embeddings of $x_t$ and $x_0$. When the baseline is BLIP, we use its ITC block to extract embeddings. For CLIP and LEMoN, we use CLIP embeddings.

**Learning Procedure Details.** Once the trajectory embeddings are constructed, they can be used as features to predict whether a given image-caption pair contains an error. While any standard classification model could be applied at this stage, we use XGBoost and CART due to their simplicity, as our primary goal is to demonstrate the effectiveness of our approach. More sophisticated models could be explored to further improve the performance gap between *TRACED* and the original baseline.

For datasets that use the Karpathy split, we combine the original training and validation sets. We then perform 3-fold grid-search cross-validation to select the best model and hyperparameters. The complete grid searches are provided in Appendix A.9. The best-performing model (XGBoost or CART) and its corresponding hyperparameters are selected based on the highest cross-validation AUC score.

## 4.4 Main Results and Analysis

**Error Detection.** The results on error detection are presented in Table 1, where accuracy scores and accuracy improvements are averaged over all applicable noise types and random seeds. *TRACED* yields consistent and significant gains over each baseline, highlighting its effectiveness for error detection.

Table 1 suggests that the Elimination algorithm often generates more informative trajectories for error detection compared to GCD and Fast GCD. We attribute this to two main factors.

- The Elimination algorithm progressively removes words from the caption, producing a trajectory in which the alignment score typically increases before decreasing. Unlike GCD and FGCD, which replace some tokens to increase alignment, Elimination reflects

Table 1: Comparison of *TRACED* with baselines. Mean accuracy and mean accuracy improvement of *TRACED* vs. baselines, averaged over 3 seeds and noise types (noun, random for MM-IMDB; noun, random, fine-grained for Flickr30k and MS-COCO at 50% noise), with standard errors.

| Dataset | Method | Algorithm | Acc. (%) | Improvement (%) |
|---|---|---|---|---|
| Flickr-30k | *TRACED-*BLIP (ITM) | Elim | **89.5 ± 0.2** | **1.3 ± 0.2** |
| | | FGCD | 89.2 ± 0.2 | 0.8 ± 0.2 |
| | | GCD | 88.8 ± 0.2 | 0.3 ± 0.2 |
| | BLIP (ITM) | - | 88.5 ± 0.3 | 0.0 ± 0.0 |
| | *TRACED-*BLIP (ITC) | Elim | 88.1 ± 0.2 | 0.9 ± 0.1 |
| | | FGCD | **88.1 ± 0.1** | **0.9 ± 0.3** |
| | | GCD | 88.0 ± 0.2 | 0.7 ± 0.1 |
| | BLIP (ITC) | - | 87.4 ± 0.3 | 0.0 ± 0.0 |
| | *TRACED-*LEMoN$_{OPT}$ | Elim | 85.6 ± 0.4 | 1.8 ± 0.2 |
| | | FGCD | 85.5 ± 0.3 | 1.9 ± 0.2 |
| | | GCD | **85.7 ± 0.4** | **2.1 ± 0.5** |
| | LEMoN$_{OPT}$ | - | 84.3 ± 0.3 | 0.0 ± 0.0 |
| | *TRACED-*LEMoN$_{FIX}$ | Elim | 85.0 ± 0.3 | 1.7 ± 0.5 |
| | | FGCD | 85.0 ± 0.3 | 1.7 ± 0.1 |
| | | GCD | **85.6 ± 0.4** | **2.6 ± 0.5** |
| | LEMoN$_{FIX}$ | - | 83.9 ± 0.3 | 0.0 ± 0.0 |
| | *TRACED-*CLIP | Elim | **85.7 ± 0.1** | **2.8 ± 0.4** |
| | | FGCD | 85.5 ± 0.1 | 2.6 ± 0.2 |
| | | GCD | 85.5 ± 0.3 | 2.5 ± 0.5 |
| | CLIP | - | 83.8 ± 0.2 | 0.0 ± 0.0 |
| MM-IMDB | *TRACED-*LEMoN$_{OPT}$ | Elim | **79.0 ± 0.0** | **1.4 ± 0.2** |
| | | FGCD | 78.0 ± 0.1 | 0.2 ± 0.1 |
| | | GCD | 78.3 ± 0.1 | 0.5 ± 0.1 |
| | LEMoN$_{OPT}$ | - | 77.9 ± 0.2 | 0.0 ± 0.0 |
| | *TRACED-*LEMoN$_{FIX}$ | Elim | **78.3 ± 0.1** | **2.4 ± 0.2** |
| | | FGCD | 77.2 ± 0.1 | 0.9 ± 0.2 |
| | | GCD | 77.6 ± 0.1 | 1.4 ± 0.2 |
| | LEMoN$_{FIX}$ | - | 76.5 ± 0.1 | 0.0 ± 0.0 |
| | *TRACED-*CLIP | Elim | **78.5 ± 0.0** | **1.8 ± 0.1** |
| | | FGCD | 77.5 ± 0.1 | 0.4 ± 0.2 |
| | | GCD | 77.8 ± 0.1 | 0.9 ± 0.0 |
| | CLIP | - | 77.2 ± 0.1 | 0.0 ± 0.0 |
| MS-COCO | *TRACED-*BLIP (ITM) | Elim | **90.5 ± 0.2** | **1.7 ± 0.1** |
| | | FGCD | 89.8 ± 0.1 | 0.9 ± 0.1 |
| | | GCD | 89.7 ± 0.1 | 0.8 ± 0.1 |
| | BLIP (ITM) | - | 89.1 ± 0.1 | 0.0 ± 0.0 |
| | *TRACED-*BLIP (ITC) | Elim | **88.7 ± 0.1** | **1.8 ± 0.0** |
| | | FGCD | 88.4 ± 0.0 | 1.4 ± 0.1 |
| | | GCD | 88.1 ± 0.1 | 1.0 ± 0.1 |
| | BLIP (ITC) | - | 87.4 ± 0.1 | 0.0 ± 0.0 |
| | *TRACED-*LEMoN$_{OPT}$ | Elim | **85.0 ± 0.3** | **1.6 ± 0.0** |
| | | FGCD | 84.6 ± 0.3 | 1.0 ± 0.1 |
| | | GCD | 84.5 ± 0.3 | 1.0 ± 0.1 |
| | LEMoN$_{OPT}$ | - | 83.8 ± 0.3 | 0.0 ± 0.0 |
| | *TRACED-*LEMoN$_{FIX}$ | Elim | **84.3 ± 0.1** | **2.3 ± 0.1** |
| | | FGCD | 83.9 ± 0.2 | 1.8 ± 0.2 |
| | | GCD | 83.9 ± 0.3 | 1.8 ± 0.2 |
| | LEMoN$_{FIX}$ | - | 82.6 ± 0.1 | 0.0 ± 0.0 |
| | *TRACED-*CLIP | Elim | **84.5 ± 0.2** | **2.5 ± 0.2** |
| | | FGCD | 83.7 ± 0.2 | 1.5 ± 0.2 |
| | | GCD | 83.8 ± 0.1 | 1.6 ± 0.1 |
| | CLIP | - | 82.7 ± 0.2 | 0.0 ± 0.0 |

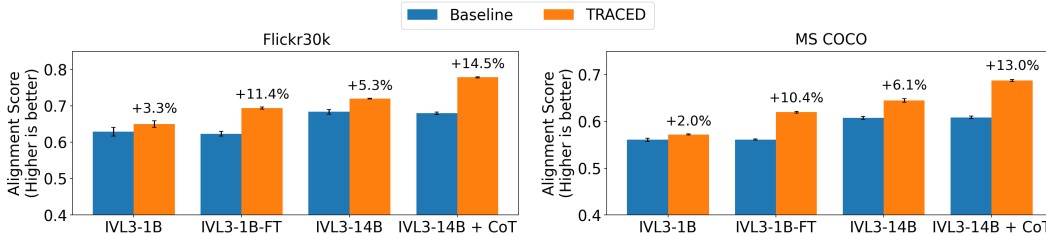

Figure 5: Impact of *TRACED* on BLIP alignment scores of the corrected captions from InternVL3-1B, InternVL3-1B-FT, and InternVL3-14B (with CoT prompting for the 14B model) on Flickr30k and MS COCO. Results are shown after five fixing steps with two fixing prompts at each step (see Appendix A.5), and averaged over three seeds with standard errors. 50% of the samples are corrupted using fine-grained noise. InternVL3 models are abbreviated as IVL3 in the plot.

both the positive and negative contributions of each individual word, showing which align with the image and which do not.

- The Elimination algorithm operates in a much more constrained search space, which introduces a form of regularization. In contrast, GCD and Fast GCD allow broader substitutions, sometimes leading to non-meaningful token replacements that nonetheless increase the alignment score.

A detailed breakdown by noise type is provided in Table 6 (Appendix A.10). The largest gains from *TRACED* occur under the fine-grained noise setting, where subtle word changes make detection especially challenging and baseline methods struggle most. Under this regime, *TRACED* achieves up to 7.5% improvement in accuracy compared to the baseline. These results highlight *TRACED* 's strength in handling more realistic and semantically nuanced errors.

**Importance of $N$, $T$, and $c$.** Appendix A.13 and Appendix A.14 analyze the effect of the number of candidate edits $N$, the trajectory length $T$, and the semantic similarity metric $c$ on the performance of *TRACED*. We find that *TRACED* is primarily sensitive to $T$: increasing $T$ lead to significant and consistent gains in accuracy. However, only a few exploration steps are typically required to locate the erroneous token (Figure 3), and small values of $T$ already achieve near-optimal performance.

**Performance of *TRACED* across caption lengths and noise levels.** Appendix A.15 reports the performance of *TRACED* as a function of caption length, showing consistent improvements across sentence lengths. Appendix A.11 further evaluates *TRACED* under varying noise levels, with results indicating consistent gains over the baseline.

**Application of *TRACED* on Error Correction.** Figure 5 shows the impact of *TRACED* on the correction performance of InternVL3-1B, including with fine-tuning (InternVL3-1B-FT) and InternVL3-14B, with and without CoT prompting. We omit CoT results for the 1B variants here, as this smaller model exhibits limited reasoning capabilities. We observe that *TRACED* consistently enhances the BLIP alignment score of the generated captions, regardless of model scale. Notably, for InternVL3-14B, performance improves further when combined with CoT prompting, suggesting that token-level error information is highly informative to optimize over the alignment metric. Our results show that our procedure can successfully optimize caption quality with respect to an alignment metric. In particular, our approach is metric-agnostic and could also benefit future scoring methods.

**Downstream captioning performance.** Appendix A.12 evaluates the impact of using *TRACED* to identify erroneous samples for filtering or correction prior to fine-tuning BLIP-2 Li et al. (2023). Both approaches improve BLIP-2's captioning performance, with correction yielding the largest gains, and *TRACED* consistently outperforming baseline methods. Overall, on Flickr30k, our procedure yields up to a 1-point improvement in BLEU-4 over training on the noisy data.

## 4.5 IMPORTANCE OF THE TRAJECTORY

To assess the importance of the caption trajectory for effective error detection, we compare the full *TRACED* trajectory to three simplified single point variants: (i) using only the first step ($s(x_0, y)$; note that $c(x_0, x_0) = 1$ provides no additional signal, (ii) using only the last step ($s(x_T, y)$ and

$c(x_T, x_0)$, and (iii) using the mean of all alignment and similarity values across the trajectory ($\frac{1}{T+1}\sum_{t=0}^{T} s(x_t, y)$ and $\frac{1}{T}\sum_{t=1}^{T} c(x_t, x_0)$). Table 2 reports the mean percent change in test accuracy for each variant, relative to using the complete trajectory.

Table 2: Mean percent improvement in Test Acc. when using only the first step, last step or mean trajectory alone in *TRACED*, compared to using the whole trajectory. Experiments are conducted on Flickr30k using the Elimination algorithm. Results are averaged over 3 seeds and all 3 noise types (50% noise), with standard errors reported.

| METHOD | FIRST STEP | LAST STEP | MEAN TRAJECTORY |
|---|---|---|---|
| BLIP (ITM) | $-1.25 \pm 0.23$ | $-43.07 \pm 0.61$ | $-4.90 \pm 0.16$ |
| BLIP (ITC) | $-0.85 \pm 0.08$ | $-40.54 \pm 0.39$ | $-5.48 \pm 0.30$ |
| LEMON$_{\text{OPT}}$ | $-1.75 \pm 0.17$ | $-38.67 \pm 0.82$ | $-5.72 \pm 0.09$ |
| LEMON$_{\text{FIX}}$ | $-1.62 \pm 0.46$ | $-38.18 \pm 0.42$ | $-5.67 \pm 0.55$ |
| CLIP | $-2.62 \pm 0.35$ | $-38.49 \pm 0.30$ | $-5.37 \pm 0.02$ |

Retaining only the first step causes the mildest decline. Using only the last step yields the sharpest drop (over 38% in all cases), and averaging over the trajectory performs slightly better but still falls far behind the full sequence. These results highlight the importance of modeling the trajectory, which captures how alignment evolves and provides richer information than a single-point summary.

## 4.6 MAXIMIZING OR MINIMIZING THE SCORING FUNCTION?

In *TRACED*, we proposed to generate the trajectories by maximizing the image-caption alignment score $s$ at each step. To test whether the opposite strategy is also effective, we compare against a variant that minimizes $s$ instead. Table 3 reports the mean percent improvement in test accuracy when using the minimization approach, relative to maximization.

Table 3: Mean percent improvement in Test Acc. when generating *TRACED*'s trajectory by minimizing $s$ rather than maximizing it. Experiments are conducted on Flickr30k using the Elimination algorithm. Results are averaged over 3 seeds and all 3 noise types (50% noise), with standard errors.

| BLIP (ITM) | BLIP (ITC) | LEMON$_{\text{FIX}}$ | LEMON$_{\text{OPT}}$ | CLIP |
|---|---|---|---|---|
| $-0.76$ $\pm 0.12$ | $-0.46$ $\pm 0.17$ | $-0.70$ $\pm 0.24$ | $-0.58$ $\pm 0.34$ | $-0.24$ $\pm 0.15$ |

Across all baselines, maximizing the alignment score yields modest but consistent improvements over minimization. This suggests that constructing trajectories toward higher-scoring captions, rather than worse ones, provides a more reliable signal for detecting inconsistencies.

## 5 CONCLUSION

We presented *TRACED*, a flexible and efficient framework for image-caption error detection. By iteratively improving captions and analyzing alignment and semantic similarity over time, *TRACED* extracts rich signals that help distinguish between correct and erroneous image-caption pairs. Our framework can be applied on top of existing error detection methods such as CLIP, LEMoN and BLIP, consistently boosting their performance across multiple datasets and noise types. We also introduced a new fine-grained noise generation process that reflects real-world annotation errors and provides a more challenging benchmark for evaluation. Beyond improving error detection, *TRACED* can be used for interpretability by revealing which parts of a caption contribute most to misalignment. We show that this token-level error information can effectively guide VLMs for caption correction, producing captions that are better aligned with their corresponding images.

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

## A  APPENDIX

### A.1  CONTRIBUTION OF IMAGE-CAPTION ALIGNMENT AND CAPTION-CAPTION SIMILARITY METRICS

To isolate the contribution of each trajectory evaluation metric, we conduct ablation studies using *TRACED* with either the alignment score $s$ or the semantic similarity score $c$ alone. Table 4 reports the mean percent change in test accuracy when using one of the two metrics alone, relative to using both jointly.

Across all baselines, using either metric in isolation results in a consistent and significant drop in performance. The alignment score $s$ alone is much more informative, likely because a notable increase in alignment often signals an error in the original caption. In contrast, the semantic similarity score $c$ is less useful on its own, as captions along the trajectory may differ substantially from the original, reducing its standalone discriminative power. However, combining $s$ and $c$ consistently yields the best performance: $s$ captures the degree of alignment improvement, while $c$ indicates whether that improvement involves a substantial semantic change or only a minor rephrasing.

Table 4: Mean percent improvement in Test Acc. when using either $s$ or $c$ alone in *TRACED*, compared to using both jointly. Experiments are conducted on MS-COCO using the Elimination algorithm. Results are averaged over 3 seeds and all 3 noise types (50% noise), with standard errors reported.

| METHOD | ALIGNMENT IMAGE-CAPTION $s(x,y)$ | SIMILARITY CAPTION-CAPTION $c(x_t, x_0)$ |
|---|---|---|
| BLIP (ITM) | $-0.41 \pm 0.13$ | $-6.03 \pm 0.11$ |
| BLIP (ITC) | $-0.59 \pm 0.12$ | $-10.18 \pm 0.16$ |
| LEMON$_{\text{OPT}}$ | $-0.55 \pm 0.12$ | $-7.22 \pm 0.17$ |
| LEMON$_{\text{FIX}}$ | $-0.41 \pm 0.21$ | $-6.44 \pm 0.33$ |
| CLIP | $-0.60 \pm 0.04$ | $-7.00 \pm 0.19$ |

## A.2 EXPLORATION ALGORITHMS DETAILS

The Elimination algorithm iteratively removes the token whose deletion increases in alignment score the most, until no tokens remain.

---

**Algorithm 2** Elimination Algorithm

---

**Input:** initial caption $x$
Note $x = (w_1, \ldots, w_L)$ with $w_1, \ldots, w_L$ the tokens in caption $x$
**for** $i = 1$ **to** $L$ **do**
   $x^{(i)} \leftarrow (w_1, \ldots, w_{i-1}, w_{i+1}, \ldots, w_L)$
**end for**
**Output:** $\{x^{(1)}, \ldots, x^{(L)}\}$

---

The Greedy Coordinate Descent (GCD) algorithm perturbs the caption by replacing individual tokens. For each position, it selects top-$K$ promising replacements based on the gradient of the alignment score. A subset of candidate captions is then generated by sampling token replacements at random.

---

**Algorithm 3** Greedy Coordinate Descent (GCD)

---

**Input:** Initial caption $x = (w_1, \ldots, w_L)$, image $y$, scoring function $s$, evaluation function $e$, number of candidates $N$, top-$K$ promising replacements per position
Let $\mathcal{V}$ be the vocabulary, and $e(v)$ the embedding of token $v \in \mathcal{V}$
**for** $j = 1$ **to** $L$ **do**
   Compute top-$K$ replacements for $w_{j_0}$:
   $\mathcal{X}_j \leftarrow \text{Top-}K \left\{ \nabla_{e(w_{j_0})} s(x,y)^T (e(v) - e(w_j)) \mid v \in \mathcal{V} \right\}$
**end for**
**for** $k = 1$ **to** $N$ **do**
   $j \sim \text{Uniform}(\{1, \ldots, L\})$
   $w'_j \sim \text{Uniform}(\mathcal{X}_j)$
   $x^{(k)} \leftarrow (w_1, \ldots, w_{j-1}, w'_j, w_{j+1}, \ldots, w_L)$
**end for**
**Output:** $\{x^{(1)}, \ldots, x^{(N)}\}$

---

This algorithm is inspired by the GCD method proposed by Zou et al. (2023), which was originally developed for adversarial attacks on large language models. In our work, we adapt this approach for the purpose of improving image captions.

The Fast GCD algorithm is a more efficient alternative to full GCD. It first applies the Elimination Algorithm to identify the token position $j_0 \in [L]$ that most negatively impacts alignment. Gradient-based substitution is then restricted to this single position. Unlike full GCD, which randomly sam-

ples $N$ captions from a pool of $K \times L$ candidates ($N \ll K \times L$), Fast GCD can exhaustively evaluate all $K$ candidate replacements at position $j_0$. This approach enables to find better token substitutions using a reduced number of forward passes through the alignment scoring function $s$.

---

**Algorithm 4** Fast Greedy Coordinate Descent (Fast GCD)

---

**Input:** Initial caption $x = (w_1, \ldots, w_L)$, image $y$, scoring function $s$, evaluation function $e$, top-$K$ promising replacements per coordinate
Let $\mathcal{V}$ be the vocabulary and $e(v)$ the embedding of token $v \in \mathcal{V}$
Run Elimination Algorithm: $\{x^{(e,1)}, \ldots, x^{(e,L)}\} \leftarrow \text{Elim}(x)$
Select most promising coordinate: $j_0 \leftarrow \arg\max_{j \in [L]} s(x^{(e,j)}, y)$
Compute top-$K$ replacements for $w_{j_0}$:
$\mathcal{X}_{j_0} \leftarrow \text{Top-}K \left\{ \nabla_{e(w_{j_0})} s(x, y)^T (e(v) - e(w_{j_0})) \mid v \in \mathcal{V} \right\}$
**for** $w' \in \mathcal{X}_{j_0}$ **do**
    $x^{(w')} \leftarrow (w_1, \ldots, w_{j_0-1}, w', w_{j_0+1}, \ldots, w_L)$
**end for**
**Output:** $\{x^{(w')} \mid w' \in \mathcal{X}_{j_0}\}$

---

### A.3 ADDITIONAL EXAMPLES OF CAPTION TRAJECTORIES

Figure 8 illustrates the behavior of *TRACED* using the Elimination Algorithm on an example with a correct caption.

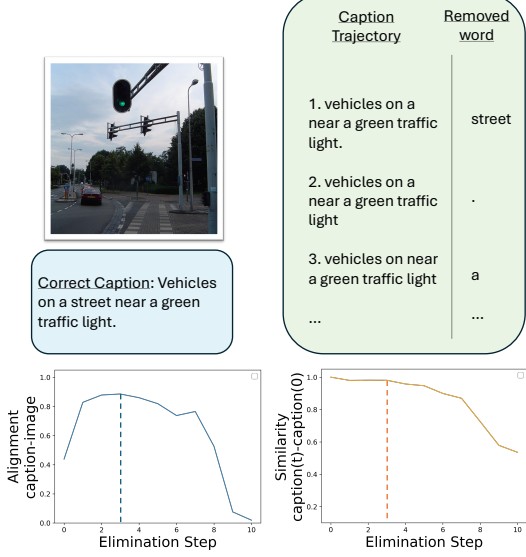

Figure 6: *TRACED* improves the BLIP-based image-caption alignment score (ITM) on an MS COCO example Lin et al. (2014), with minimal semantic change in the revised captions, suggesting the original pair is likely accurate.

We then show in Figures 7 and 8 the trajectories obtained with GCD and Fast GCD on the examples in Figures 3 and 6 respectively.

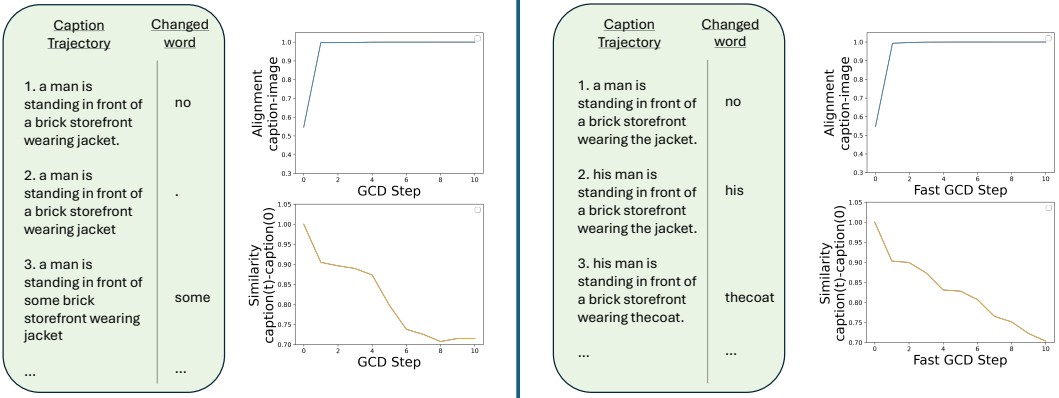

Figure 7: Caption trajectories using GCD (left) and Fast GCD (right) for the example in Figure 3. In both cases, *TRACED* identifies "no" as the source of misalignment and further improves the caption's alignment with the image.

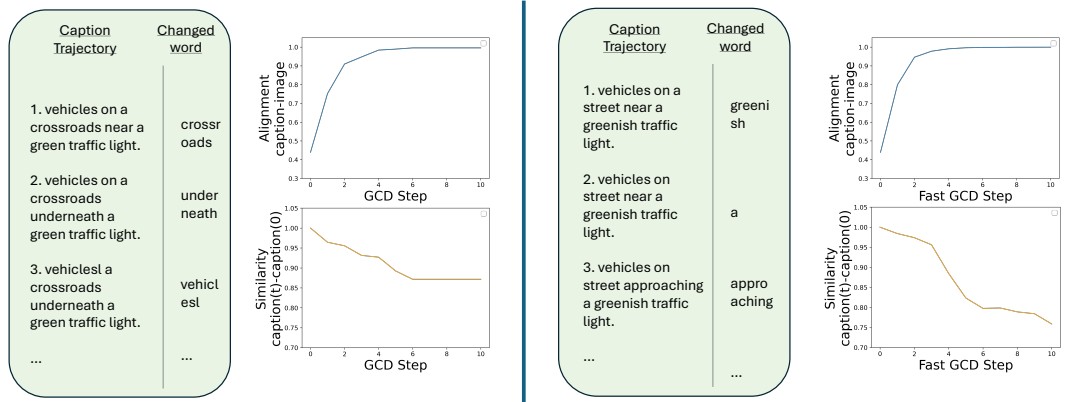

Figure 8: Caption trajectories using GCD (left) and Fast GCD (right) for the example in Figure 6. In both cases, *TRACED* improves the caption's alignment with the image using only minor semantic edits.

### A.4 NEW BENCHMARK DATASET CREATION

Prior work on error detection in image captioning introduces noise via full caption swaps as a means of constructing evaluation benchmarks (Zhang et al., 2025). However, such swaps replace the entire caption, often resulting in text unrelated to the original. In contrast, real-world annotation errors can be more subtle, with annotators correctly describing an image but misrepresenting specific details. To better capture fine-grained noise, we propose a new approach for constructing a challenging benchmark by modifying only a few words within each caption. More specifically, for each original caption, we leverage a large language model to generate $K = 20$ variants that maintain the same structure but introduce small semantic errors. The exact prompt is provided in Appendix A.18.

While many generated options are useful, some may be too similar to the initial caption. To filter these, we apply Alignscore (Zha et al., 2023), a factual consistency metric based on a fine-tuned natural language inference model. Alignscore assigns low scores to captions that either omit key information or contradict the original caption. The selected variants thus differ meaningfully in content while remaining structurally close, effectively modeling fine-grained semantic noise. To increase variability across seeds, we select the 2 least-aligned sentences for each sample.

Figure 9 illustrates this generation and filtering process.

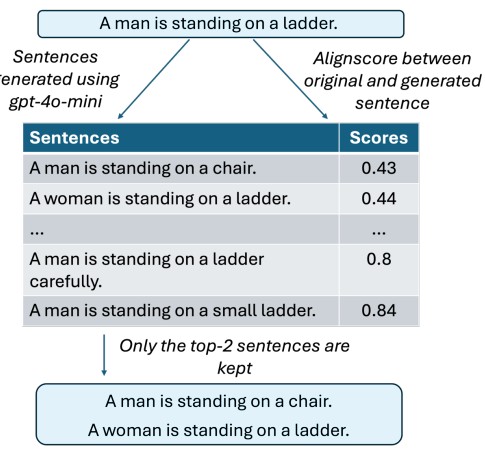

## A.5 CAPTION CORRECTION HYPERPARAMETERS

Figure 4 outlines the general caption correction framework used in this paper, where *TRACED* is applied to improve the alignment metric of corrected sentences and fix the potential errors. The way trajectory information is provided, however, can vary. In our experiments, we find that repeating the same prompt across multiple iterations often improves performance, as the model progressively refines the caption. We also observe that asking the model first to fix the caption without any guidance (to avoid bias from our signals) and then with our token-level error information appears to be the best strategy.

Figure 9: Fine-grained noise generation pipeline. Given an original caption, an LLM generates 20 variants. Alignscore then evaluates the factual consistency of each variant with respect to the original. The least aligned (lowest-scoring) sentences are selected as fine-grained noisy captions.

The main hyperparameters of our correction framework are:

- The number of fixing steps: how many times the procedure of error detection followed by error correction is applied. At each step, new flagged words are obtained with *TRACED*.

- The number of correction attempts per fixing step, denoted as $k$: at each fixing step, how many times the VLM is asked to revise the caption (using the same flagged words).

- The correction strategy: the manner in which token-level information from *TRACED* is provided to the VLM.

In our experiments, we evaluate the following correction strategies:

- Without *TRACED*: the model is asked $k$ times to correct the caption without any guidance from our interpretable error detection framework. In this case, the prompt in Figure 13 is used with [1] empty.

- With *TRACED* only: the model receives token-level error information from our framework and is asked to fix the caption with this informtion $k$ times. Here, the prompt in Figure 13 is used with [1] containing guidance on the errors in the caption.

- Without then with *TRACED*: the model performs $k-1$ correction steps without token-level guidance ([1] empty), followed by a final step where *TRACED* provides token-level error information ([1] filled).

Figure 10 highlights the impact of correction strategies. Supplying token-level error information to the VLM in a direct, naive way can sometimes perform worse than providing no guidance at all. In contrast, with an improved strategy, both the 1B and 14B models are able to leverage the information provided to generate captions that align more closely with the image. Interestingly, we find that only two prompts are often sufficient to consistently surpass the baseline which doesn't use *TRACED*. This is why we show results for two fixing iterations (Without–then–with *TRACED* (2) and Without *TRACED* only (2)) after five fixing steps in Figure 5.

## A.6 FINE-TUNING INTERNVL3-1B.

To construct a fine-tuning dataset for InternVL3-1B, we apply the correction procedure described in Section 3.4 on the train and validation sets of MS COCO and Flickr30k using InternVL3-14B, with 50% fine-grained noise. For MS COCO, we sub-sample the train set and keep the first 30% (27,594 samples).

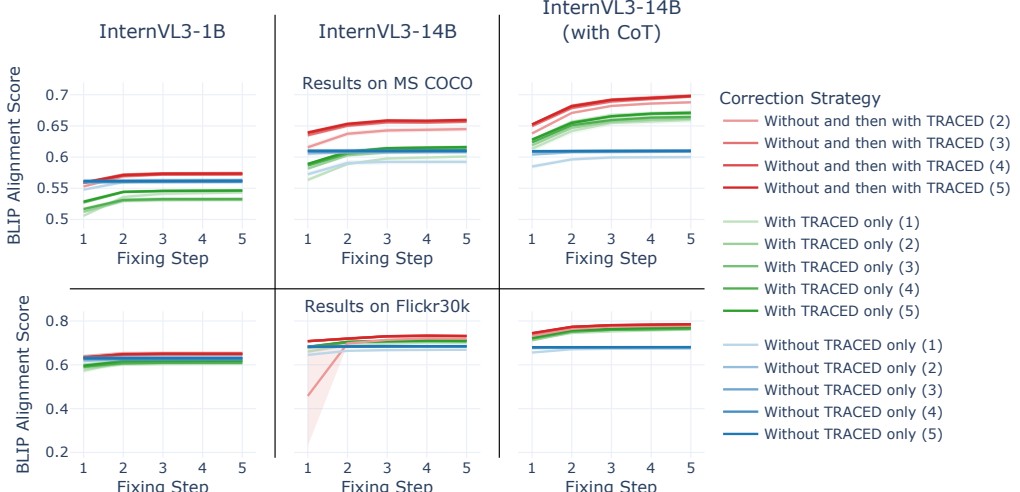

Figure 10: Caption correction results for InternVL3-1B and InternVL3-14B (with and without CoT prompting) across fixing steps. We compare three correction strategies: (1) without *TRACED*, (2) with *TRACED* only, and (3) without *TRACED* for $k-1$ iterations followed by one iteration with *TRACED*. Numbers in parentheses indicate the number of times $k$ the VLM is asked to fix the caption at each fixing step. BLIP alignment scores (ITM block) on the corrected sentences are averaged over 3 seeds with standard errors.

For each dataset and seed, InternVL3-1B is trained using cross-entropy loss (next token prediction) for five epochs, and the best checkpoint is selected based on validation correction performance (measured as the number of errors detected by *TRACED*-BLIP-ITM). The same hyperparameters (learning rate, optimizer, etc.) as in the original InternVL3 paper were used (Zhu et al., 2025). This model is denoted InternVL3-1B-FT.

## A.7 PARALLELIZATION BENEFITS

*TRACED* applies the trajectory creation and evaluation from Algorithm 1 independently to each sentence in the dataset. This enables efficient large-scale parallelization. Our method benefits from both intra-GPU and multi-GPU parallelism: given access to $n$ GPUs, the dataset can be split into $n$ subsets processed in parallel, with each GPU handling batches of samples.

## A.8 COMPUTATION OVERHEAD

Table 5 reports the computation time required by *TRACED* and the original baselines to process 1,000 image-caption pairs on a single L40 GPU. Baseline models such as BLIP, LEMoN, and CLIP are very fast as they require only a single forward pass per pair. Despite performing multiple model evaluations to construct trajectories, all variants of *TRACED*, including Elimination, Fast GCD, and GCD, remain practical and scalable. Among the proposed methods, Elimination is the most efficient, offering substantial speed advantages while maintaining among the best performance. Fast GCD achieves a strong balance between speed and trajectory quality. For example, when scaled to 1,000,000 image-caption pairs using BLIP (ITM), the most expensive baseline, and 4 L40 GPUs, Elimination completes in approximately 6.5 hours and Fast GCD takes about 2.6 days.

As described in Appendix A.7, thanks to the high degree of parallelism in our method, leveraging more GPUs can substantially further reduce total processing time.

We want to emphasize that *TRACED* needs to be applied only once to identify and filter out incorrect image-caption pairs. This one-time computational cost is reasonable for generating a cleaner dataset that can be reused across various downstream tasks, including pre-training, fine-tuning, and evaluation.

Table 5: Computation time comparison across algorithms. Reported times (in seconds) corresponds to the duration required to process 1,000 sentences with a single L40 GPU, including both trajectory exploration and alignment score evaluation.

| METHOD | ALGORITHM | COMPUTATION TIME (S) |
|---|---|---|
| BLIP (ITM) | - | $3.82 \pm 0.11$ |
| *TRACED*-BLIP (ITM) | ELIMINATION | $92.53 \pm 1.16$ |
| | FAST GCD | $905.22 \pm 0.99$ |
| | GCD | $1617.06 \pm 0.13$ |
| BLIP (ITM) | - | $3.56 \pm 0.24$ |
| *TRACED*-BLIP (ITM) | ELIMINATION | $49.05 \pm 0.28$ |
| | FAST GCD | $389.19 \pm 0.51$ |
| | GCD | $688.90 \pm 0.55$ |
| LEMoN$_{OPT}$ | - | $3.13 \pm 0.08$ |
| *TRACED*-LEMoN$_{OPT}$ | ELIMINATION | $43.77 \pm 0.59$ |
| | FAST GCD | $451.28 \pm 0.43$ |
| | GCD | $799.03 \pm 1.79$ |
| LEMoN$_{FIX}$ | - | $3.19 \pm 0.40$ |
| *TRACED*-LEMoN$_{FIX}$ | ELIMINATION | $43.44 \pm 0.44$ |
| | FAST GCD | $452.48 \pm 1.07$ |
| | GCD | $802.13 \pm 1.76$ |
| CLIP | - | $2.40 \pm 0.07$ |
| *TRACED*-CLIP | ELIMINATION | $43.10 \pm 0.57$ |
| | FAST GCD | $444.80 \pm 0.55$ |
| | GCD | $788.97 \pm 0.29$ |

## A.9 HYPERPARAMETERS

**Trajectory Generation Hyperparameters.** Depending on the exploration strategy, the caption trajectory generation from Algorithm 1 involves a few hyperparameters:

- Elimination Algorithm: We set $T = L$ and $N = \frac{L(L-1)}{2}$, where $L$ is the caption length. The algorithm removes one token at a time, selecting the one whose removal most improves the alignment score $s$, and continues until there is no token in the sentence.
- GCD Algorithm: We use $T = 10$, $K = 128$, and $N = 256$.
- Fast GCD Algorithm: We set $T = 10$, $k = 128$ and $N = K = 128$ since we explore all $K$ promising replacements for the single token identified via Elimination Algorithm.

**Grid Searches.** The hyperparameter grids used for model selection are as follows:

XGBoost hyperparameters:

- `max_depth` $\in \{3, 4, 5\}$
- `learning_rate` $\in \{0.01, 0.05, 0.1, 0.5\}$
- `n_estimators` $\in \{50, 100, 200, 400\}$

CART hyperparameters:

- `max_depth` $\in \{1, 5, 10, +\infty\}$

## A.10 RESULTS PER NOISE TYPE

We present in Table 6 the impact of *TRACED* on various baselines across the three noise types we evaluate. *TRACED* consistently improves performance across all baselines and noise settings. Notably, the gains are more substantial for noise types that are harder to detect. For example, improvements are modest for random noise, where baselines already achieve over 97% Accuracy on

Flickr30k and MS COCO. On the contrary, improvements are much more pronounced on the Fine-Grained noise and on MM-IMDb, which present more challenging errors for the existing methods.

Table 6: Comparison of *TRACED* with baselines. "Elim" and "FGCD" denote Elimination and Fast GCD, respectively. Results are averaged over 3 seeds for each noise type (50% noise). We report mean accuracy and mean accuracy improvement compared to the baseline, with standard errors.

| DATASET | METHOD | ALG. | RANDOM | | NOUN | | FINE-GRAINED | |
|---|---|---|---|---|---|---|---|---|
| | | | ACC. (%) | IMPROVEMENT (%) | ACC. (%) | IMPROVEMENT (%) | ACC. (%) | IMPROVEMENT (%) |
| FLICKR-30K | *TRACED*-BLIP (ITM) | ELIM | 98.2 ± 0.1 | 0.4 ± 0.1 | 93.8 ± 0.3 | 0.2 ± 0.2 | **76.6 ± 0.3** | **3.3 ± 0.8** |
| | | FGCD | **98.3 ± 0.1** | **0.5 ± 0.0** | **93.8 ± 0.3** | **0.3 ± 0.1** | 75.4 ± 0.4 | 1.7 ± 0.7 |
| | | GCD | 98.1 ± 0.2 | 0.3 ± 0.1 | 93.6 ± 0.3 | 0.0 ± 0.0 | 74.5 ± 0.5 | 0.5 ± 0.7 |
| | BLIP (ITM) | - | 97.8 ± 0.1 | 0.0 ± 0.0 | 93.6 ± 0.3 | 0.0 ± 0.0 | 74.2 ± 0.8 | 0.0 ± 0.0 |
| | *TRACED*-BLIP (ITC) | ELIM | 97.8 ± 0.1 | 0.1 ± 0.0 | **93.4 ± 0.2** | **1.1 ± 0.4** | 73.1 ± 0.4 | 1.4 ± 0.3 |
| | | FGCD | **97.9 ± 0.1** | **0.2 ± 0.1** | 93.0 ± 0.3 | 0.7 ± 0.2 | **73.3 ± 0.3** | **1.9 ± 0.7** |
| | | GCD | 97.8 ± 0.1 | 0.1 ± 0.1 | 93.1 ± 0.5 | 0.6 ± 0.4 | 73.0 ± 0.0 | 1.4 ± 0.5 |
| | BLIP (ITC) | - | 97.7 ± 0.1 | 0.0 ± 0.0 | 92.5 ± 0.5 | 0.0 ± 0.0 | 72.0 ± 0.4 | 0.0 ± 0.0 |
| | *TRACED*-LEMON$_{OPT}$ | ELIM | **97.5 ± 0.1** | **0.0 ± 0.0** | 90.8 ± 0.1 | 1.8 ± 0.4 | 68.5 ± 1.0 | 3.6 ± 0.8 |
| | | FGCD | 97.5 ± 0.1 | -0.0 ± 0.0 | 89.7 ± 0.1 | 0.6 ± 0.3 | 69.4 ± 0.9 | 5.0 ± 0.9 |
| | | GCD | 97.3 ± 0.0 | -0.2 ± 0.1 | 90.0 ± 0.3 | 0.9 ± 0.3 | **69.8 ± 1.0** | **5.5 ± 1.3** |
| | LEMON$_{OPT}$ | - | 97.5 ± 0.1 | 0.0 ± 0.0 | 89.2 ± 0.3 | 0.0 ± 0.0 | 66.1 ± 0.7 | 0.0 ± 0.0 |
| | *TRACED*-LEMON$_{FIX}$ | ELIM | **97.7 ± 0.1** | **0.5 ± 0.1** | 89.7 ± 0.2 | 0.4 ± 0.2 | 67.7 ± 0.7 | 4.1 ± 1.4 |
| | | FGCD | 97.1 ± 0.2 | -0.1 ± 0.1 | 89.5 ± 0.3 | 0.2 ± 0.1 | 68.4 ± 0.6 | 5.1 ± 0.4 |
| | | GCD | 97.0 ± 0.2 | -0.1 ± 0.2 | **90.0 ± 0.3** | **0.8 ± 0.2** | **69.8 ± 0.8** | **7.2 ± 1.3** |
| | LEMON$_{FIX}$ | - | 97.2 ± 0.0 | 0.0 ± 0.0 | 89.3 ± 0.4 | 0.0 ± 0.0 | 65.1 ± 0.6 | 0.0 ± 0.0 |
| | *TRACED*-CLIP | ELIM | **97.6 ± 0.1** | **0.4 ± 0.1** | **90.7 ± 0.2** | **1.6 ± 0.4** | 68.9 ± 0.4 | 6.2 ± 1.2 |
| | | FGCD | 97.3 ± 0.1 | 0.1 ± 0.1 | 89.4 ± 0.1 | 0.1 ± 0.2 | **69.7 ± 0.3** | **7.5 ± 0.7** |
| | | GCD | 97.1 ± 0.2 | -0.1 ± 0.2 | 89.9 ± 0.3 | 0.7 ± 0.0 | 69.4 ± 0.4 | 7.0 ± 1.4 |
| | CLIP | - | 97.2 ± 0.1 | 0.0 ± 0.0 | 89.3 ± 0.3 | 0.0 ± 0.0 | 64.8 ± 0.6 | 0.0 ± 0.0 |
| MM-IMDB | *TRACED*-LEMON$_{OPT}$ | ELIM | **81.4 ± 0.3** | **1.3 ± 0.1** | **76.6 ± 0.3** | **1.5 ± 0.5** | - | - |
| | | FGCD | 80.6 ± 0.3 | 0.3 ± 0.1 | 75.5 ± 0.3 | 0.1 ± 0.2 | - | - |
| | | GCD | 80.8 ± 0.4 | 0.6 ± 0.2 | 75.8 ± 0.3 | 0.4 ± 0.4 | - | - |
| | LEMON$_{OPT}$ | - | 80.3 ± 0.3 | 0.0 ± 0.0 | 75.4 ± 0.2 | 0.0 ± 0.0 | - | - |
| | *TRACED*-LEMON$_{FIX}$ | ELIM | **80.9 ± 0.2** | **2.0 ± 0.1** | **75.7 ± 0.1** | **2.8 ± 0.4** | - | - |
| | | FGCD | 80.0 ± 0.1 | 0.8 ± 0.1 | 74.4 ± 0.0 | 1.0 ± 0.5 | - | - |
| | | GCD | 80.1 ± 0.1 | 1.0 ± 0.2 | 75.0 ± 0.3 | 1.8 ± 0.4 | - | - |
| | LEMON$_{FIX}$ | - | 79.3 ± 0.2 | 0.0 ± 0.0 | 73.6 ± 0.3 | 0.0 ± 0.0 | - | - |
| | *TRACED*-CLIP | ELIM | **80.9 ± 0.3** | **1.5 ± 0.1** | **76.1 ± 0.2** | **2.1 ± 0.1** | - | - |
| | | FGCD | 80.1 ± 0.2 | 0.5 ± 0.2 | 74.9 ± 0.1 | 0.4 ± 0.2 | - | - |
| | | GCD | 80.3 ± 0.1 | 0.7 ± 0.2 | 75.3 ± 0.1 | 1.0 ± 0.2 | - | - |
| | CLIP | - | 79.7 ± 0.3 | 0.0 ± 0.0 | 74.6 ± 0.2 | 0.0 ± 0.0 | - | - |
| MS-COCO | *TRACED*-BLIP (ITM) | ELIM | **98.9 ± 0.2** | **0.4 ± 0.0** | **92.1 ± 0.1** | **0.5 ± 0.0** | **80.4 ± 0.4** | **4.4 ± 0.1** |
| | | FGCD | 98.8 ± 0.0 | 0.4 ± 0.1 | 92.1 ± 0.1 | 0.4 ± 0.0 | 78.6 ± 0.5 | 2.0 ± 0.3 |
| | | GCD | **98.9 ± 0.1** | 0.4 ± 0.1 | 92.1 ± 0.1 | 0.4 ± 0.1 | 78.2 ± 0.3 | 1.5 ± 0.2 |
| | BLIP (ITM) | - | 98.5 ± 0.1 | 0.0 ± 0.0 | 91.7 ± 0.1 | 0.0 ± 0.0 | 77.0 ± 0.2 | 0.0 ± 0.0 |
| | *TRACED*-BLIP (ITC) | ELIM | **98.7 ± 0.0** | **0.2 ± 0.1** | **90.8 ± 0.1** | **0.9 ± 0.1** | **76.8 ± 0.4** | **4.4 ± 0.1** |
| | | FGCD | 98.5 ± 0.1 | 0.0 ± 0.0 | 90.2 ± 0.1 | 0.2 ± 0.1 | 76.5 ± 0.3 | 4.1 ± 0.4 |
| | | GCD | 98.6 ± 0.1 | 0.1 ± 0.1 | 90.3 ± 0.2 | 0.2 ± 0.1 | 75.5 ± 0.3 | 2.6 ± 0.2 |
| | BLIP (ITC) | - | 98.5 ± 0.1 | 0.0 ± 0.0 | 90.0 ± 0.0 | 0.0 ± 0.0 | 73.6 ± 0.5 | 0.0 ± 0.0 |
| | *TRACED*-LEMON$_{OPT}$ | ELIM | **97.8 ± 0.1** | **0.1 ± 0.1** | **86.3 ± 0.3** | **1.4 ± 0.2** | **70.8 ± 0.6** | **3.2 ± 0.1** |
| | | FGCD | 97.7 ± 0.1 | 0.0 ± 0.1 | 85.5 ± 0.4 | 0.4 ± 0.1 | 70.4 ± 0.6 | 2.6 ± 0.2 |
| | | GCD | 97.7 ± 0.1 | -0.0 ± 0.1 | 85.2 ± 0.5 | 0.1 ± 0.1 | 70.7 ± 0.6 | 3.1 ± 0.1 |
| | LEMON$_{OPT}$ | - | 97.7 ± 0.1 | 0.0 ± 0.0 | 85.1 ± 0.4 | 0.0 ± 0.0 | 68.7 ± 0.5 | 0.0 ± 0.0 |
| | *TRACED*-LEMON$_{FIX}$ | ELIM | 97.7 ± 0.1 | 0.0 ± 0.0 | **85.6 ± 0.2** | **2.9 ± 0.3** | 69.6 ± 0.4 | 3.9 ± 0.5 |
| | | FGCD | **97.8 ± 0.1** | **0.1 ± 0.0** | 84.6 ± 0.3 | 1.8 ± 0.3 | 69.3 ± 0.7 | 3.4 ± 0.6 |
| | | GCD | 97.7 ± 0.1 | 0.1 ± 0.1 | 84.2 ± 0.3 | 1.3 ± 0.5 | **69.7 ± 0.6** | **3.9 ± 0.2** |
| | LEMON$_{FIX}$ | - | 97.7 ± 0.1 | 0.0 ± 0.0 | 83.1 ± 0.5 | 0.0 ± 0.0 | 67.1 ± 0.6 | 0.0 ± 0.0 |
| | *TRACED*-CLIP | ELIM | **97.7 ± 0.1** | **0.2 ± 0.0** | **85.8 ± 0.2** | **2.3 ± 0.2** | **69.9 ± 0.4** | **4.9 ± 0.2** |
| | | FGCD | 97.5 ± 0.1 | -0.0 ± 0.1 | 84.6 ± 0.1 | 0.9 ± 0.3 | 69.0 ± 0.6 | 3.6 ± 0.4 |
| | | GCD | 97.6 ± 0.1 | 0.1 ± 0.0 | 84.4 ± 0.2 | 0.7 ± 0.1 | 69.3 ± 0.4 | 4.0 ± 0.3 |
| | CLIP | - | 97.5 ± 0.1 | 0.0 ± 0.0 | 83.8 ± 0.3 | 0.0 ± 0.0 | 66.6 ± 0.3 | 0.0 ± 0.0 |

## A.11 PERFORMANCE OF *TRACED* ACROSS NOISE LEVELS

The main results of the paper (Table 1 and Table 6) are reported for a noise ratio of 50%. In this section, we evaluate the performance of *TRACED* across additional noise levels—10%, 20%, 30%, and 40%. Because these settings introduce class imbalance, we report Test AUC scores in Table 7. Across all noise levels, *TRACED* consistently outperforms the baselines. As seen previously, the

| Noise type | Method | 10% | 20% | 30% | 40% |
|---|---|---|---|---|---|
| Fine-grained | *TRACED*-BLIP (ITM) | $83.5 \pm 1.5$ | $82.6 \pm 0.8$ | $84.0 \pm 1.3$ | $84.4 \pm 0.8$ |
| | BLIP (ITM) | $80.8 \pm 1.0$ | $79.1 \pm 0.6$ | $80.6 \pm 1.1$ | $81.4 \pm 0.7$ |
| | *TRACED*-LEMoN$_{OPT}$ | $76.4 \pm 1.1$ | $75.4 \pm 2.1$ | $75.0 \pm 0.7$ | $74.7 \pm 1.0$ |
| | LEMoN$_{OPT}$ | $73.5 \pm 2.1$ | $70.7 \pm 0.9$ | $71.4 \pm 0.8$ | $72.1 \pm 1.5$ |
| | *TRACED*-CLIP | $76.7 \pm 1.4$ | $76.0 \pm 1.8$ | $75.4 \pm 1.0$ | $75.8 \pm 1.1$ |
| | CLIP | $71.5 \pm 2.5$ | $70.0 \pm 1.5$ | $70.7 \pm 1.3$ | $70.6 \pm 1.2$ |
| Noun | *TRACED*-BLIP (ITM) | $99.1 \pm 0.3$ | $98.8 \pm 0.2$ | $98.8 \pm 0.2$ | $98.5 \pm 0.2$ |
| | BLIP (ITM) | $99.0 \pm 0.3$ | $98.8 \pm 0.2$ | $98.8 \pm 0.2$ | $98.5 \pm 0.2$ |
| | *TRACED*-LEMoN$_{OPT}$ | $96.2 \pm 0.5$ | $95.9 \pm 0.7$ | $96.0 \pm 0.3$ | $95.1 \pm 0.2$ |
| | LEMoN$_{OPT}$ | $95.7 \pm 0.5$ | $95.6 \pm 0.8$ | $95.3 \pm 0.4$ | $94.5 \pm 0.1$ |
| | *TRACED*-CLIP | $96.4 \pm 0.3$ | $96.3 \pm 0.4$ | $96.2 \pm 0.3$ | $95.5 \pm 0.2$ |
| | CLIP | $95.5 \pm 0.3$ | $95.6 \pm 0.7$ | $95.4 \pm 0.3$ | $94.7 \pm 0.3$ |
| Random | *TRACED*-BLIP (ITM) | $99.7 \pm 0.2$ | $99.6 \pm 0.1$ | $99.8 \pm 0.1$ | $99.8 \pm 0.1$ |
| | BLIP (ITM) | $99.6 \pm 0.3$ | $99.7 \pm 0.1$ | $99.7 \pm 0.1$ | $99.7 \pm 0.1$ |
| | *TRACED*-LEMoN$_{OPT}$ | $99.5 \pm 0.1$ | $99.6 \pm 0.2$ | $99.5 \pm 0.1$ | $99.5 \pm 0.1$ |
| | LEMoN$_{OPT}$ | $99.4 \pm 0.1$ | $99.5 \pm 0.2$ | $99.3 \pm 0.3$ | $99.4 \pm 0.1$ |
| | *TRACED*-CLIP | $99.3 \pm 0.2$ | $99.6 \pm 0.1$ | $99.6 \pm 0.2$ | $99.6 \pm 0.1$ |
| | CLIP | $99.3 \pm 0.2$ | $99.5 \pm 0.2$ | $99.5 \pm 0.1$ | $99.5 \pm 0.1$ |

Table 7: Comparison of *TRACED*, using the Elimination Algorithm, against baseline models across noise levels (10%, 20%, 30%, and 40%) on Flickr30k. Test AUCs are averaged over three seeds on with standard errors.

gains are especially substantial under fine-grained noise, where baseline methods tend to struggle the most.

## A.12 DOWNSTREAM CAPTIONING PERFORMANCE

In this section, we evaluate the impact of our filtering and correction procedure on the downstream performance of a captioning model, BLIP-2 Li et al. (2023). In particular, we fine-tune BLIP-2 on (i) the Flickr30k datasets containing 50% fine-grained noise and (ii) cleaned datasets obtained by removing sentences flagged as incorrect by BLIP (the strongest baseline) or by *TRACED*-BLIP. In addition to filtering, we also test the correction strategy, where sentences predicted as wrong by *TRACED*-BLIP are replaced with corrected versions produced either by InternVL3-1B alone or using the token-level correction procedure described in Section 3.4.

We fine-tune BLIP-2 using LoRA (rank = 4) and report BLEU-4, ROUGE, and CIDEr scores. We train the models for 3 epochs, with a learning rate of 1e-5, AdamW and using early stopping. The table below summarizes the results:

| Method | BLEU-4 (%) | ROUGE (%) | CIDEr (%) |
|---|---|---|---|
| Noisy | $31.5 \pm 0.2$ | $56.5 \pm 0.1$ | $69.2 \pm 0.4$ |
| Filtering with BLIP | $30.8 \pm 0.5$ | $56.1 \pm 0.3$ | $67.9 \pm 1.3$ |
| Filtering with TRACED-BLIP | $31.8 \pm 0.1$ | $56.6 \pm 0.1$ | $69.6 \pm 0.2$ |
| Ideal Filtering | $31.8 \pm 0.1$ | $56.8 \pm 0.1$ | $70.2 \pm 0.4$ |
| Correction with InternVL3-1B | $32.0 \pm 0.1$ | $56.8 \pm 0.0$ | $70.2 \pm 0.4$ |
| Correction with InternVL3-1B + TRACED | $32.5 \pm 0.1$ | $56.9 \pm 0.1$ | $70.4 \pm 0.5$ |

Table 8: Impact of filtering and correction, using either the baseline or *TRACED*, on the downstream caption quality of BLIP-2. In both correction settings, corrections are applied to the same samples predicted as erroneous by *TRACED*-BLIP, isolating the effect of *TRACED* 's interpretable tokens in the correction pipeline.

Filtering with *TRACED*-BLIP achieves performance very close to Ideal Filtering and substantially outperforms the baseline (BLIP-filtering) across all metrics. Applying caption correction to the samples flagged as erroneous by *TRACED*-BLIP yields additional gains in BLEU-4, ROUGE, and CIDEr, and incorporating *TRACED* 's interpretable tokens into the correction process further boosts downstream caption quality. Overall, our *TRACED*-based correction pipeline improves BLIP-2's captioning performance by up to 1 point in BLEU-4, 0.4 in ROUGE, and 1.2 in CIDEr compared to training on the noisy dataset.

## A.13  IMPACT OF $N$ AND $T$ ON *TRACED*

Algorithm 1 relies on two key hyperparameters: $N$, the number of candidate edits considered at each step, and $T$, the trajectory length. In this section, we analyze the sensitivity of *TRACED* to these hyperparameters.

**Impact of $N$.** For the Elimination algorithm, $N$ is inherently small: by construction it matches the number of tokens in the caption and decreases by one at each iteration, making exploration progressively faster. Consequently, we focus our analysis on the role of $N$ in the GCD algorithm, and report in Figure 11 the impact of varying $N$ on performance on Flickr30k.

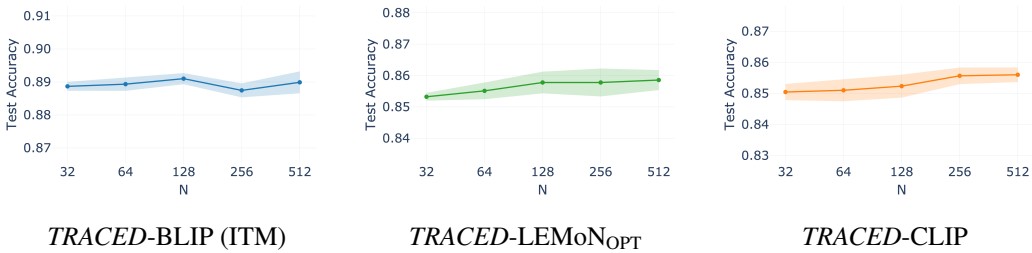

| *TRACED*-BLIP (ITM) | *TRACED*-LEMoN$_{OPT}$ | *TRACED*-CLIP |

Figure 11: Impact of varying $N$ on the performance of *TRACED* when applied to BLIP (ITM), LEMoN$_{OPT}$, and CLIP using the GCD algorithm. Mean accuracies on Flickr30k over 3 seeds and noise types (noun, random and fine-grained at 50% noise), are reported with standard errors.

**Impact of $T$.** We evaluated the influence of trajectory length to determine how much information is gained from longer versus shorter paths. We used the Elimination algorithm to conduct this experiment. The results on Flickr30k are presented in Figure 12. We observe that larger values of $T$ generally lead to improved performance, particularly for CLIP and LEMoN$_{OPT}$. However, a single step of Elimination ($T = 1$) is often sufficient to achieve near-optimal, and sometimes even optimal, performance. This behavior can be explained by the fact that in many cases, only one or a few exploratory steps are required for our procedure to identify an incorrect token along the caption trajectory (see Figure 3) and therefore determine whether the caption is correct or not.

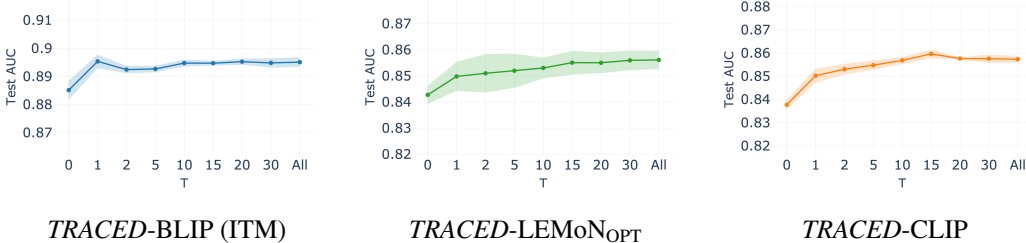

| *TRACED*-BLIP (ITM) | *TRACED*-LEMoN$_{OPT}$ | *TRACED*-CLIP |

Figure 12: Impact of varying $T$ on the performance of *TRACED* (Elimination algorithm) when applied to BLIP (ITM), LEMoN$_{OPT}$, and CLIP. Mean accuracies on Flickr30k over 3 seeds and noise types (noun, random and fine-grained at 50% noise), are reported with standard errors.

## A.14  IMPORTANCE OF THE SEMANTIC SIMILARITY SCORE

We additionally study the impact of the semantic similarity function $c$ on the performance of *TRACED*. For BLIP (ITM), we compared using the ITC component (as described in Section 4.3)

with using CLIP. For both LEMoN$_{OPT}$ and CLIP, we performed the other substitution and evaluated whether replacing CLIP with BLIP (ITC) affected performance. The resulting comparisons are provided below.

| $s$ | Noise Type | $c = $ CLIP | $c = $ BLIP (ITC) |
|---|---|---|---|
| BLIP (ITM) | fine-grained | $76.0 \pm 0.2$ | $76.6 \pm 0.3$ |
| | noun | $93.7 \pm 0.3$ | $93.8 \pm 0.3$ |
| | random | $98.0 \pm 0.1$ | $98.2 \pm 0.1$ |
| LEMoN$_{OPT}$ | fine-grained | $68.5 \pm 1.0$ | $69.2 \pm 1.0$ |
| | noun | $90.8 \pm 0.1$ | $90.5 \pm 0.2$ |
| | random | $97.5 \pm 0.1$ | $97.4 \pm 0.3$ |
| CLIP | fine-grained | $68.9 \pm 0.4$ | $70.0 \pm 0.2$ |
| | noun | $90.7 \pm 0.2$ | $90.9 \pm 0.3$ |
| | random | $97.6 \pm 0.1$ | $97.5 \pm 0.3$ |

Table 9: Impact of changing $c$ on the performance of *TRACED*. Mean accuracies on Flickr30k for each noise type are computed over 3 seeds with standard errors.

While the choice of BLIP (ITC) seems preferable for $s = $ BLIP (ITM), the choice of $c$ does not seem to have a decisive impact on the final performance, and both CLIP and BLIP (ITC) appear to be effective metrics $c$ for error-detection.

## A.15 PERFORMANCE OF *TRACED* ACROSS CAPTION LENGTH

To assess whether the effect of *TRACED* on the baseline depends on caption length, we compute test accuracies for both *TRACED* and the baselines across caption-length bins on MS COCO. The corresponding results are shown in Table 10. We find that *TRACED* consistently improves upon the different baselines across all caption-length ranges, indicating that the trajectory-guided approach is beneficial regardless of the caption length.

| Noise type | Method | (6.999, 9.0] | (9.0, 10.0] | (10.0, 11.0] | (11.0, 14.0] | (14.0, 34.0] |
|---|---|---|---|---|---|---|
| Fine-grained | *TRACED*-BLIP (ITM) | $80.5 \pm 0.5$ | $80.9 \pm 1.2$ | $80.1 \pm 0.5$ | $80.2 \pm 0.2$ | $79.1 \pm 0.4$ |
| | BLIP | $76.5 \pm 0.3$ | $78.3 \pm 0.9$ | $76.8 \pm 0.7$ | $77.8 \pm 0.4$ | $74.4 \pm 2.0$ |
| | *TRACED*-LEMoN$_{OPT}$ | $70.5 \pm 0.9$ | $70.4 \pm 0.9$ | $71.1 \pm 0.5$ | $71.2 \pm 0.5$ | $71.3 \pm 1.5$ |
| | LEMON_OPT | $68.6 \pm 0.7$ | $68.5 \pm 1.0$ | $68.0 \pm 1.3$ | $69.6 \pm 1.0$ | $68.1 \pm 1.1$ |
| | *TRACED*-CLIP | $70.3 \pm 0.5$ | $68.9 \pm 0.3$ | $69.8 \pm 1.2$ | $70.3 \pm 0.6$ | $68.5 \pm 1.3$ |
| | CLIP | $66.7 \pm 0.4$ | $66.8 \pm 0.9$ | $65.0 \pm 0.8$ | $68.1 \pm 0.3$ | $65.4 \pm 2.0$ |
| Noun | *TRACED*-BLIP (ITM) | $91.9 \pm 0.3$ | $92.6 \pm 0.1$ | $92.6 \pm 0.1$ | $91.6 \pm 0.3$ | $91.7 \pm 0.6$ |
| | BLIP | $91.7 \pm 0.4$ | $91.8 \pm 0.1$ | $92.1 \pm 0.2$ | $90.9 \pm 0.4$ | $92.1 \pm 0.9$ |
| | *TRACED*-LEMoN$_{OPT}$ | $86.0 \pm 0.5$ | $86.3 \pm 0.7$ | $86.4 \pm 0.8$ | $86.6 \pm 0.2$ | $87.8 \pm 1.2$ |
| | LEMON_OPT | $84.1 \pm 0.3$ | $85.4 \pm 1.1$ | $85.6 \pm 0.7$ | $85.8 \pm 0.2$ | $88.3 \pm 1.1$ |
| | *TRACED*-CLIP | $85.6 \pm 0.3$ | $86.2 \pm 1.0$ | $85.6 \pm 0.7$ | $85.3 \pm 0.0$ | $87.3 \pm 2.1$ |
| | CLIP | $83.1 \pm 0.2$ | $83.9 \pm 0.7$ | $84.3 \pm 0.4$ | $84.5 \pm 0.7$ | $86.0 \pm 1.2$ |
| Random | *TRACED*-BLIP (ITM) | $98.8 \pm 0.0$ | $98.8 \pm 0.4$ | $99.1 \pm 0.2$ | $99.0 \pm 0.2$ | $98.5 \pm 0.5$ |
| | BLIP | $98.3 \pm 0.1$ | $98.7 \pm 0.3$ | $98.4 \pm 0.4$ | $98.5 \pm 0.3$ | $98.5 \pm 0.5$ |
| | *TRACED*-LEMoN$_{OPT}$ | $97.7 \pm 0.1$ | $98.3 \pm 0.1$ | $97.8 \pm 0.1$ | $97.4 \pm 0.2$ | $98.7 \pm 0.2$ |
| | LEMON_OPT | $97.6 \pm 0.2$ | $98.1 \pm 0.0$ | $97.7 \pm 0.0$ | $97.5 \pm 0.2$ | $98.0 \pm 0.2$ |
| | *TRACED*-CLIP | $97.7 \pm 0.3$ | $98.2 \pm 0.0$ | $97.6 \pm 0.2$ | $97.3 \pm 0.2$ | $98.0 \pm 0.2$ |
| | CLIP | $97.5 \pm 0.3$ | $98.1 \pm 0.1$ | $97.4 \pm 0.1$ | $97.2 \pm 0.1$ | $98.0 \pm 0.2$ |

Table 10: Comparison of *TRACED*, using the Elimination Algorithm, with baseline models across caption-length bins on MS COCO. The bins correspond to the (0%, 25%], (25%, 50%], (50%, 75%], (75%, 95%], and (95%, 100%] percentiles of caption length. For each bin, test accuracies are averaged over three seeds, with standard errors reported.

## A.16 EXAMPLES OF ERRORS WITH FINE-GRAINED NOISE

We show below a list of examples of noisy captions generated as described in A.4.

**Wrong object/element**

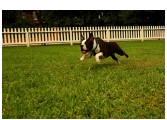

**True:** A black and white dog is running through the grass.
**Noisy:** A black and white cat is running through the grass.

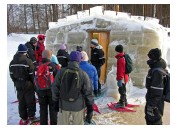

**True:** Several students waiting outside an igloo.
**Noisy:** Several students waiting outside a car.

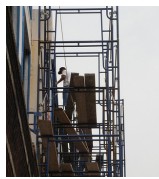

**True:** A man in a white shirt stands high up on scaffolding.
**Noisy:** A woman in a red shirt stands high up on scaffolding.

**Wrong negation**

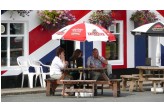

**True:** Three people are sitting at an outside picnic bench with an umbrella.
**Noisy:** Two people are sitting at an inside picnic table without an umbrella.

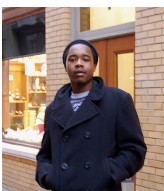

**True:** A man is standing in front of a brick storefront wearing a black jacket.
**Noisy:** A man is standing in front of a brick storefront wearing no jacket.

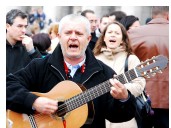

**True:** A man playing an acoustic guitar and singing with a group of people behind him including a woman who is singing along.
**Noisy:** A man not playing any instrument and speaking with a group of people behind him including a woman who is taking notes.

**Wrong action**

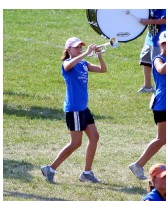

**True** Girl wearing blue shirt and black shorts plays trumped outside.
**Noisy:** The girl wearing a blue shirt and black shorts sings outside.

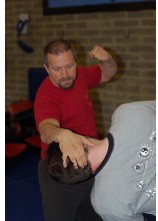

**True:** One man holds another man's head down and prepares to punch him in the face.
**Noisy:** One man holds another man's head down while his friends cheer him on.

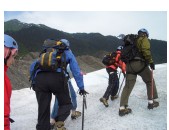

**True:** A group of people are hiking up an icy hillside.
**Noisy:** A group of people are skiing down an icy hillside.

**Wrong number**

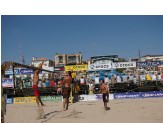

**True:** A group of spectators watch a men's sand volleyball game.
**Noisy:** A solitary spectator watches a men's sand volleyball game.



**True:** Shaft of light in a cave shows three spelunkers.
**Noisy:** A shaft of light in a cave reveals one spelunker.

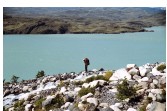

**True:** Person standing on rocky edge of water with hilly land in background.
**Noisy:** A group of people standing on the sandy beach with flat land in the background.

Table 11: Examples of true versus noisy captions with fine-grained noise from our new benchmark dataset, derived from Flickr30k, illustrating errors in the objects, negations, actions and quantities.

**Wrong object/element**

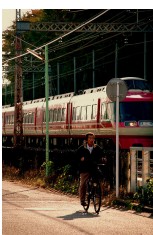

**True:** a guy that is riding his bike next to a train
**Noisy:** A guy that is riding his bike next to a bus.

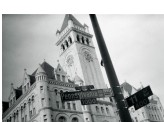

**True:** An old black and white photo of Pennsylvania Avenue.
**Noisy:** An old black and white photo of Fifth Avenue.

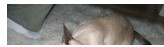

**True:** a cat that is eating some kind of banana
**Noisy:** A dog that is eating some kind of banana.

**Wrong negation**

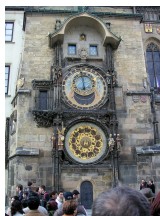

**True:** Many people gather around a building with clocks
**Noisy:** Many people gather around a building without clocks.

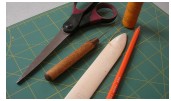

**True:** A pencil is sitting on a ruler with a pair of scissors.
**Noisy:** A pencil is sitting on a ruler without any scissors.

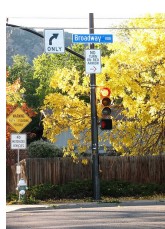

**True:** a pole that has a sign on it
**Noisy:** A pole that has no sign on it.

**Wrong action**

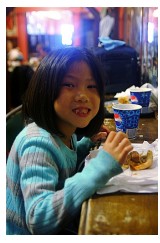

**True** Little girl smiles for the camera as she eats her sandwich
**Noisy:** Little girl smiles for the camera as she drinks her juice.

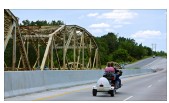

**True:** A couple riding a motorcycle down a street.
**Noisy:** A couple walking down a street.

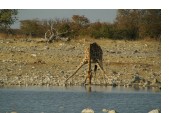

**True:** a giraffe bending down to drink from a body of water
**Noisy:** A giraffe stretching its neck to eat from a tree.

**Wrong number**

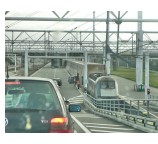

**True:** a photo of a train heading down the tracks
**Noisy:** A photo of multiple trains heading down the tracks.

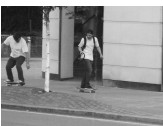

**True:** Two friends are skateboarding down the street to their next destination.
**Noisy:** Three friends are skateboarding down the street to their next destination.

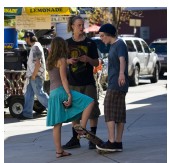

**True:** a group of three people talking to each other on the sidewalk with a skateboard
**Noisy:** A group of four people talking to each other on the sidewalk with a bicycle.

Table 12: Examples of true versus noisy captions with fine-grained noise from our new benchmark dataset, derived from MS COCO, illustrating errors in the objects, negations, actions and quantities.

**Wrong color**

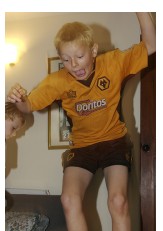

**True:** A boy wearing a orange Doritos jersey jumps up in the air.
**Noisy:** A girl wearing a blue Doritos jersey jumps up in the air.

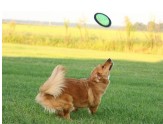

**True:** A brown dog about to catch a green Frisbee.
**Noisy:** A black dog about to catch a red Frisbee.

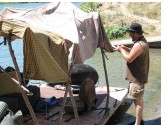

**True:** A man in brown building a raft.
**Noisy:** A man in red building a raft.

**Distributed errors**

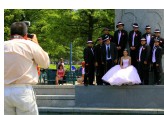

**True:** A bride in a light pink dress poses for a picture with male relatives and is being photographed by a man in a cream shirt with white pants.
**Noisy:** A bride in a light pink dress poses for a picture with her siblings and is being photographed by a girl in a green shirt with a floral skirt.

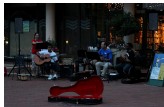

**True:** A band is of four members including a woman and three men are playing their instruments with an open guitar case in front of them.
**Noisy:** An orchestra is of ten members including a woman and nine men are playing their instruments with an open violin case in front of them.

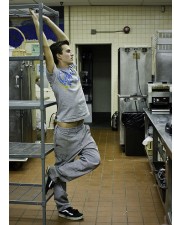

**True:** A young man in a gray tee-shirt and gray sweatpants stands by a metal tiered shelf in an industrial kitchen, holding the top edge of the metal structure, with one leg resting on the knee of the other leg.
**Noisy:** A young woman in a blue tee-shirt and black sweatpants stands by a metal tiered shelf in an industrial kitchen, holding the bottom edge of the metal structure, with one leg resting on the knee of the other leg.

Table 13: Examples of true versus noisy captions with fine-grained noise from our new benchmark dataset, derived from Flickr30, illustrating color errors and distributed errors.

**Wrong color**

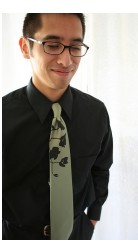

**True:** A young man wearing black attire and a flowered tie is standing and smiling.
**Noisy:** A young man wearing blue attire and a flowered tie is standing and smiling.

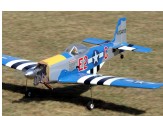

**True:** A small blue plane sitting on top of a field.
**Noisy:** A large red plane sitting on top of a field.

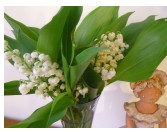

**True:** white flowers in a vase with arranged leaves
**Noisy:** Red flowers in a vase with arranged leaves.

**Distributed errors**

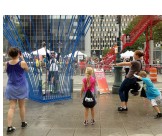

**True:** Woman taking a picture of someone standing behind a sculpture and a child pushing another woman towards the sculpture.
**Noisy:** A woman taking a picture of a dog standing behind a sculpture and a child pushing another woman away from the sculpture.

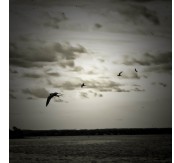

**True:** Several birds that are flying together over a body of water.
**Noisy:** A flock of ravens that are cawing together above a forest.

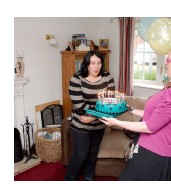

**True:** A woman holding a blue birthday cake with stars and candles on it and another woman in front of the cake.
**Noisy:** A woman holding a green birthday cake with stars and no candles on it and another woman in front of the cake.

Table 14: Examples of true versus noisy captions with fine-grained noise from our new benchmark dataset, derived from MS COCO, illustrating color errors and distributed errors.

### A.17 PROMPT FOR CAPTION CORRECTION

We present the prompts used for caption correction in Section 3.4. To isolate the effect of *TRACED*, we employ identical prompts across settings, with the only modification being the addition of a sentence specific to our framework ([1] in Figures 13 and 14). For instance, in Figure 4, [1] is replaced with: "The following words are wrong: 'a', 'player', 'female', 'a', 'rack', 'is', '.', 'the'. Other words might be wrong too."

```
<|image|>
### Task:
You are an expert caption editor.

Please check the caption for factual or visual accuracy. [1]

If the caption is inaccurate, rewrite it using only the original words or minimal substitutions to make it accurate.

**Avoid adding new details or descriptions. Keep the revised caption short and concise.**

If the caption is accurate, return it exactly as is. Don't make any changes.

Respond with only the corrected caption, enclosed in quotation marks.

### Caption:
"A female tennis player is on the court with a racket."
### Corrected Caption:
```

Figure 13: Prompt used for caption correction. In the case of *TRACED*, token-level error information is provided in [1].

```
<|image|>
### Task:
You are an expert caption editor.

Please check the caption for factual or visual accuracy. [1]

If the caption is inaccurate, rewrite it using only the original words or minimal substitutions to make it accurate.

**Avoid adding new details or descriptions. Keep the revised caption short and concise.**

If the caption is accurate, return it exactly as is. Don't make any changes.

Think step by step, explain why there is one or multiple errors, and then provide the corrected caption, enclosed in quotation marks. Don't output anything after the corrected sentence.

### Caption:
"A female tennis player is on the court with a racket."
```

Figure 14: Prompt used for caption correction using Chain-of-Thought prompting. In the case of *TRACED*, token-level error information is provided in [1].

## A.18    PROMPT FOR FINE-GRAINED NOISE TYPE

We use the prompts in Figures 15 and 16 to generate 20 candidate noisy captions for each caption in the MS COCO (Lin et al., 2014) and Flickr30k (Plummer et al., 2015) datasets.

---

# Sentence Variation Generator

For a given input sentence, generate up to 20 variations that have similar structure but convey clearly different meanings. Follow these systematic modification rules:

## Analysis Requirements
1. First, identify the basic structure of the sentence
2. Identify all components: subject, predicate, object (if any), attributives (if any), adverbials (if any), and clauses (if any)
3. Create variations by modifying one or two components per variation

## Component Modification Guidelines

### Subject Modifications (1-2 variations)
- Change the quantity of the subject: e.g., "A man" → "Two men"; "A group of people" → "One person"
- Change the subject itself: e.g., "A man" → "A woman"; "A person" → "An animal"; "A group of students" → "A group of police officers"

**Examples:**
- Original: "The doctor examined the patient carefully."
  - Variation: "The nurse examined the patient carefully." (Changed subject identity)
  - Variation: "Several doctors examined the patient carefully." (Changed subject quantity)

### Predicate Modifications (1-2 variations)
- Replace the verb with an unrelated verb: e.g., "standing" → "sitting"; "waving" → "running"
- Ensure the object (if present) is also modified to fit the new verb context

**Examples:**
- Original: "The chef prepared a delicious meal for the guests."
  - Variation: "The chef served a delicious meal for the guests." (Changed verb)
  - Variation: "The chef ruined a delicious meal for the guests." (Changed verb to opposite meaning)

### Object Modifications (1-2 variations)
- Replace the noun in the object with a different noun: ensure it still fits the context but differs significantly from the original
- If there is an object complement, modify it to express an opposite or completely different meaning

**Examples:**
- Original: "She bought a new car with her bonus."
  - Variation: "She bought a new house with her bonus." (Changed object noun)
  - Variation: "She bought an old car with her bonus." (Changed object attribute to opposite)

### Attributive Modifications (1-2 variations)
- For adjectives or nouns serving as attributives, replace with contextually appropriate words that convey completely different meanings
- For numerical attributives, change the quantity
- For prepositional phrases or infinitives, modify to maintain context while expressing significantly different meaning

**Examples:**
- Original: "The tall building on the corner was recently renovated."
  - Variation: "The historic building on the corner was recently renovated." (Changed attributive adjective)
  - Variation: "The tall building in the downtown area was recently renovated." (Changed attributive prepositional phrase)

---

Figure 15: First part of the prompt used to create the fine-grained noise using `gpt-4o-mini`.

## A.19    USE OF LLM

LLMs were used to polish the writing of this paper.

### Adverbial Modifications (1-2 variations)
- For time and place adverbials, change to completely different times or locations
- For manner and degree adverbials, change the adverb to its antonym or to a completely different adverb
- For reason, result, condition adverbials, modify the corresponding clause

**Examples:**
- Original: "They quickly finished their homework before dinner."
  - Variation: "They slowly finished their homework before dinner." (Changed manner adverbial to opposite)
  - Variation: "They quickly finished their homework after midnight." (Changed time adverbial)

### Clause Modifications (1-2 variations)
- Identify the components within the clause and modify them according to the guidelines above

**Examples:**
- Original: "She said that she would come to the party if she finished her work."
  - Variation: "She said that she would skip the party if she finished her work." (Changed predicate in the clause)
  - Variation: "She said that she would come to the party unless she finished her work." (Changed condition in the adverbial clause)

## Important Requirements

1. Each variation should differ from the original in 1-2 components only
2. Modifications must be significant enough to clearly change the meaning of the sentence
3. The modified sentence must maintain grammatical correctness and contextual coherence
4. If the original sentence is too short to generate 20 variations, provide as many as reasonably possible
5. Consider the context of the sentence and ensure modifications are contextually appropriate
6. Number each variation sequentially (1-20)

## Output Format
1. [Modified sentence 1]
2. [Modified sentence 2]
...
20. [Modified sentence 20]

Original: {sentence}

Figure 16: Second part of the prompt used to create the fine-grained noise using `gpt-4o-mini`.

