# OpenReview forum: "Seeing What’s Wrong: A Trajectory-Guided Approach to Caption Error Detection"
_ICLR.cc/2026/Conference — ICLR 2026 Poster_

### Official Review · Reviewer_PQfZ · 2025-10-27

**Soundness:** 3
**Presentation:** 3
**Contribution:** 3
**Rating:** 6
**Confidence:** 3

**Summary:**

This paper proposes TRACED, a new method for caption error detection. By looking at caption trajectories – iterative edit traces starting from the original caption – TRACED looks for atomic edits that greatly improve an image-text alignment score, finding that these frequently correspond to fine-grained misalignments between the original caption and image. Results show that TRACED is effective at identifying and correcting caption errors, potentially complementing existing methods.

**Strengths:**

The idea of using caption trajectories to identify fine-grained misalignments is clever and appears novel.

The framework is model-agnostic, making it flexible and potentially applicable to a wide range of settings and improvements in vision-language models.

The results show a modest but significant improvement in error detection using TRACED relative to baseline methods.

Overall the paper is clearly written.

**Weaknesses:**

The evaluation seems partly circular, as the results in Sec 4.4 show that TRACED, which is designed based on single-token edits, improves error detection for synthetic edits which may be largely single-token based (L367–374). In addition TRACED relies on an automatic alignment score using models which may be biased towards the types of errors generated synthetically. This seems partly in line with Table 4 where improvements for random caption substitution are mostly very small.

There are some evident limitations of the types of trajectories used which are not clearly discussed. Conceptually, since VLMs like CLIP and BLIP struggle with compositionality, it is not clear that they will be robust to fine-grained compositional misalignments (e.g. “a jacket” vs. “no jacket”). [1] Since trajectories only use single token edits, it is not clear how or if TRACED could handle misalignments due to multi-token words for phrases.

Optimization-based edits lead to non-fluent captions, as seen in Fig 6–8 (“vehicles on near …”, “thecoat”, “vehiclesl” etc.), making it unclear whether the trajectory signal is fully meaningful. Flagged words include function words (L931, e.g. “is”). Limitations and/or robustness to these aspects should be explicitly discussed.

The improvements in error detection seem modest (Tab 1), raising questions about whether it is worth the added computational overhead. While the paper claims the method is “efficient” in several places, some of the runtimes reported in Tab 3 are rather high (although I acknowledge that this is a one-time cost).

[1] Yuksekgonul et al. When and why vision-language models behave like bags-of-words, and what to do about it? ICLR 2023

**Questions:**

How are flagged words (as in Fig 4) determined? Is it a fixed threshold applied to changes in s(...) or c(...)?

Other works explore caption filtering, recaptioning, or error correction showing improvement on downstream VLM performance. Have you tested this? How does your method compare, or is it complementary to these approaches?

Is the interpretability application (Sec 3.3) purely impressionistic, or have you tested that TRACED flagged errors correlate statistically with human judgements of the source of misalignments?

It seems like Elimination is usually more effective than the more complex GCD-based methods while being much more efficient. What is the motivation for testing GCD? Could it be improved to be competitive or does it have other conceptual or practical benefits?

In Tab 3, Fast GCD is approximately x240 slower than elimination, vs. L1608 where it is about x9 slower. Is this correct?

---

> ### Author Response · Authors · 2025-11-22
> **TRACED can handle muli-token errors and its edits are often meaningful. The computational cost is reasonable, and downstream captioning performance improves with TRACED.**
>
> We thank the reviewer for their careful reading of the paper and comments.
>
> ## 1 and 2: Ability of TRACED to handle multi-token errors
>
> TRACED is indeed strong on cases with small, localized errors, and we highlight such examples because they are among the hardest for the baselines. However, the method is not limited to single-token edits: it can remove multiple erroneous tokens over successive steps and handle more complex errors. While improvements are smaller for the random noise type, this setting is much simpler for the  the baselines (accuracies (edited on nov 26th) up to 97.8\% and 98.5\% on Flickr30k and COCO), leaving less room for improvements. In more challenging settings, including MM-IMDb and the noun noise type where entire captions are swapped and many tokens are wrong, TRACED shows clearer gains (up to 2.0\% and 2.9\% Accuracy (edited on nov 26th) respectively).
>
> ## 3. Meaningful edits for error detection
>
> We agree that this point deserves a clearer discussion in the paper. Our exploration strategies (Elimination, GCD, and FastGCD) can indeed produce captions that are not semantically or grammatically correct. This is not an issue for error detection, where the task is purely classification. This observation is precisely what motivates our approach in Section 3.4, where we leverage the flagged tokens for caption correction and rely on VLMs, models that excel at text generation, to produce fluent and coherent sentences informed by our token-level insights. We will make this motivation more explicit in the paper.
>
> We believe the intermediate captions still convey a strong and meaningful signal: the tokens that are removed or replaced earliest are often the erroneous ones, as illustrated in Figures 3, 6, 7, and 8.
>
> ## 4. Computational Efficiency
>
> While we believe that the results in Table 1 are quite promising, Table 4 shows that the improvements of our framework are consistent across settings, and are all the more important in cases where the baselines struggle most. It is worth noting that Table 1 reports averages over three noise types; this averaging dampens the effect of our gains on the fine-grained noise for instance.
>
> We do believe that the computational overhead of the Elimination Algorithm is reasonable. TRACED-BLIP can process 1,000,000 image–caption pairs in 6.5 hours using 4 L40 GPUs. If computational cost is a concern and interpretability is not required, users can use a small value of $T$, instead of eliminating all the tokens, and still achieve strong improvements over the baseline, as demonstrated in our ablation study on $T$ for Reviewer QGRZ.
> Because our procedure is highly parallelizable, with each image–caption pair processed independently, the runtime can be reduced further with additional GPUs. Finally, as the reviewer noted, the detection step is a one-time cost: once completed, the resulting cleaned dataset can be reused for downstream tasks without additional computation.
>
> ## Questions:
>
> ## 1. Token-flagging criterion used for correction
>
> For the correction task, we treat as flagged any token whose removal strictly increases the image–caption alignment score. We agree that this criterion may flag some tokens incorrectly. Although more refined flagging mechanisms could reduce such errors, our intent was to include all potentially problematic tokens in the prompt. Despite its simplicity, this strategy improves the alignment scores of the corrected captions, indicating that the flagged tokens carry meaningful information.
>
> ## 2. Downstream captioning performance improve with TRACED
>
> We believe our approach is complementary to these existing works: better filtering [1] and better captions [2] generally lead to improved captioning performance for models trained on cleaner datasets. We agree with the reviewer that evaluating downstream performance using our framework can be valuable for the reader. We conducted some experiments, described in the fourth point of our response to Reviewer ZqLU, and observed gains (on a single seed) in the downstream captioning performance of BLIP-2 when trained on datasets improved by our filtering and correction procedure. We will extend these experiments to additional seeds and include the results in the paper.
>
> [1] Zhang, H. et al. (2025). LEMoN: Label Error Detection using Multimodal Neighbors. In ICML 2025.
>
> [2] Li, J. et al. (2022). BLIP: Bootstrapping language–image pre-training for unified vision–language understanding and generation. In ICML 2022.

---

> ### Author Response · Authors · 2025-11-22
> **Evaluation of TRACED-flagged tokens**
>
> ## 3. Evaluation of TRACED-flagged tokens
>
> We thank the reviewer for the question. We conducted a quantitative analysis comparing TRACED-flagged tokens (using BLIP) with the ground-truth modified tokens in the fine-grained Flickr30k setting (by comparing the noisy captions with the original ones). Tokens are considered as incorrect if they were not present in the original clean caption.
>
> As pointed out by the reviewer, thresholds $(\delta_s, \delta_c)$ can reduce over-flagging. We report results across different $(\delta_s, \delta_c)$ to show their impact.
>
> On average across three seeds, TRACED detects at least one incorrect token in 88.8\% of examples at $(\delta_s,\delta_c)=(0, \text{none})$, and remains above 75-80\% for most thresholds. When requiring that at least 50\% of all modified tokens be detected, accuracies remain in the 60-80\% range depending on thresholds.
>
> **Accuracy of detecting at least one modified token**
>
> |  $\delta_s$ \  $\delta_c$  | 0.000| 0.001| 0.005| 0.010| 0.020|
> |--|---|---|---|---|---|
> | none| 0.888±0.009  | 0.853±0.014  | 0.823±0.017  | 0.803±0.018  | 0.780±0.009  |
> | 0.000  | 0.878±0.009  | 0.844±0.017  | 0.813±0.020  | 0.793±0.021  | 0.769±0.014  |
> | -0.001 | 0.877±0.009  | 0.841±0.016  | 0.813±0.020  | 0.793±0.021  | 0.769±0.014  |
> | -0.005 | 0.870±0.012  | 0.836±0.018  | 0.808±0.021  | 0.787±0.021  | 0.765±0.015  |
> | -0.010 | 0.855±0.014  | 0.827±0.019  | 0.801±0.022  | 0.780±0.021  | 0.758±0.014  |
> | -0.020 | 0.817±0.016  | 0.799±0.020  | 0.773±0.019  | 0.757±0.020  | 0.739±0.014  |
>
> **Accuracy of detecting at least 50% of modified tokens**
>
> | $\delta_s$ \  $\delta_c$ | 0.000| 0.001| 0.005| 0.010| 0.020|
> |--|---|---|---|---|---|
> | none| 0.801±0.011  | 0.744±0.007  | 0.695±0.002  | 0.659±0.001  | 0.636±0.004  |
> | +0.000 | 0.771±0.010  | 0.721±0.008  | 0.673±0.008  | 0.638±0.007  | 0.615±0.003  |
> | -0.001 | 0.765±0.013  | 0.720±0.009  | 0.671±0.006  | 0.637±0.007  | 0.613±0.004  |
> | -0.005 | 0.736±0.010  | 0.697±0.011  | 0.655±0.008  | 0.627±0.009  | 0.603±0.005  |
> | -0.010 | 0.711±0.007  | 0.679±0.007  | 0.637±0.006  | 0.614±0.007  | 0.589±0.006  |
> | -0.020 | 0.651±0.006  | 0.628±0.009  | 0.591±0.009  | 0.579±0.010  | 0.559±0.006  |
>
> **Proportion of flagged tokens per caption**
>
> | $\delta_s$ \  $\delta_c$ | 0.000| 0.001| 0.005| 0.010| 0.020|
> |--|---|---|---|---|---|
> | none| 0.416±0.004  | 0.348±0.004  | 0.298±0.005  | 0.268±0.004  | 0.234±0.003  |
> | 0.000  | 0.315±0.004  | 0.262±0.004  | 0.223±0.005  | 0.199±0.004  | 0.174±0.003  |
> | -0.001 | 0.307±0.004  | 0.258±0.004  | 0.220±0.005  | 0.197±0.004  | 0.173±0.003  |
> | -0.005 | 0.264±0.004  | 0.230±0.004  | 0.202±0.005  | 0.183±0.004  | 0.162±0.003  |
> | -0.010 | 0.225±0.004  | 0.201±0.004  | 0.180±0.004  | 0.164±0.004  | 0.147±0.003  |
> | -0.020 | 0.179±0.003  | 0.165±0.003  | 0.151±0.002  | 0.141±0.002  | 0.128±0.002  |
>
> These results indicate that TRACED-flagged tokens correlate strongly with actual sources of misalignment.
>
> We also provide examples illustrating that the largest changes in alignment score $s$ and semantic similarity $c$ typically occur precisely at the erroneous words:
>
> ### Example 1
>
> **True:**
> *A black and white dog is running through the grass.*
>
> **Noisy:**
> *A black and white cat is running through the grass.*
>
> | Token| Δs| Δc|
> |-----------|--------|---------|
> | cat  | 0.34| -0.11  |
> | a | 0.38| -0.03  |
> | through| 0.11| -0.03  |
> | is| 0.03| -0.01  |
> | . | 0.01| -0.00  |
> | white| 0.02| -0.21  |
>
> ---
>
> ### Example 2
>
> **True:**
> *Five people wearing winter clothing, helmets, and ski goggles stand outside in the snow.*
>
> **Noisy:**
> *Five people wearing winter clothing, helmets, and ski goggles stand inside by the fire.*
>
> | Token| Δs| Δc|
> |---------|--------|---------|
> | fire| 0.84| -0.13  |
> | inside  | 0.01| -0.05  |
> | people  | 0.00| -0.01  |
> | and| 0.00| -0.02  |
>
> ---
>
> ### Example 3
>
> **True:**
> *Several students waiting outside an igloo.*
>
> **Noisy:**
> *Several students waiting outside a school.*
>
> | Token| Δs| Δc|
> |-----------|--------|---------|
> | school| 0.10| -0.06  |
> | students  | 0.20| -0.15  |
> | . | 0.10| 0.01|
>
> ---
>
> ### Example 4
>
> **True:**
> *A man in a white shirt stands high up on scaffolding.*
>
> **Noisy:**
> *A man in a white shirt stands low down on the ground.*
>
> | Token| Δs| Δc|
> |-----------|--------|---------|
> | ground| 0.24| -0.04  |
> | low  | 0.37| -0.05  |
> | stands| 0.13| -0.07  |
> | a | 0.05| -0.02  |

---

> ### Author Response · Authors · 2025-11-22
> **Clarifications on GCD-based explorations and computational times**
>
> ## 4. Motivation for including GCD-based exploration
>
> We agree that Elimination appears stronger than GCD. However, we believe that GCD-based token replacement is a natural alternative that readers might consider, so we included it for completeness.
>
> ## 5. Computational time comparison
>
> Thank you for carefully reading the paper. We believe you are referring to Line 1068, and if so, we do think that the numbers are aligned. In the case of BLIP ITM, the Elimination strategy takes 92.53s to process 1,000 sentences on a single L40 GPU, against 905.22s for the Fast GCD Algorithm. We believe this 9.78x ratio aligns with the numbers Line 1068 (6.5 hours against 2.6 days).

---

### Official Review · Reviewer_TbST · 2025-10-30

**Soundness:** 2
**Presentation:** 3
**Contribution:** 2
**Rating:** 6
**Confidence:** 4

**Summary:**

The paper proposes **TRACED**, a model-agnostic framework for detecting erroneous image–caption pairs by analyzing a **trajectory** of edits applied to the caption that monotonically seek to improve an image–text relevance score $s$. Instead of trusting a single similarity score, TRACED constructs a sequence of candidate captions via token deletions and/or replacements, tracks how (s) evolves, and also measures semantic drift relative to the original caption. A classifier is then trained on trajectory features to predict whether the original pair is correct. The authors evaluate TRACED on MS-COCO, Flickr30k, and MM-IMDb, showing consistent AUC gains over CLIP, BLIP, and LEMoN baselines; they also introduce a “fine-grained” noise process (minimal GPT-4o-mini edits) to stress subtle errors. Beyond detection, the trajectory highlights token sources of error; these token-level hints can guide a VLM to produce higher-scoring corrected captions.

**Strengths:**

1. Moving from a single score to **trajectory statistics** is intuitive and broadly compatible with popular alignment functions (CLIP cosine, BLIP ITC/ITM, LEMoN). The paper’s Algorithm-1-style procedure and the use of both alignment evolution and semantic similarity are well-motivated.
2. Trajectories naturally localize problematic tokens (e.g., removing “no” resolves a negation error). This is a useful diagnostic for dataset cleaning pipelines and for prompting downstream caption correction.
3. Supplying token-level “flagged words” to a VLM improves BLIP alignment of corrected captions (especially with larger models + CoT prompting), supporting the claim that trajectories provide actionable signals, not just detection.

**Weaknesses:**

1. TRACED *optimizes and evaluates* using alignment metrics like BLIP ITM and CLIP similarity. While convenient, this risks **circularity**: if trajectories explicitly maximize (s), improved AUC and “correction quality” may partly reflect over-optimization of the very metric used for labels/evaluation rather than true semantic correctness.
2. A large part of the evaluation hinges on **synthetic** perturbations, including the proposed fine-grained GPT-4o-mini edits. While this nicely induces subtle errors, it is still simulated. It remains unclear how well TRACED generalizes to **real, organically noisy** web-scale datasets.
3. We’re told using both alignment and semantic-similarity signals beats using either alone, and that Elimination often wins; however, a fuller **ablation** would help isolate what matters most: trajectory length (T), candidate count (N), semantic similarity choice, and the classifier architecture.
4. While the paper argues scalability, **single-pass** baselines are very cheap; TRACED adds multiple forward passes per pair. The reported 6.5h/1M pairs (BLIP-ITM + Elimination on 4×L40) is reasonable, but not trivial for very large corpora; clearer guidance on **cost/quality trade-offs** (when Elimination suffices vs. when Fast GCD pays off) would aid adoption.

**Questions:**

N/A

---

> ### Author Response · Authors · 2025-11-22
> **Alignment score improvements, real-world evaluation and evaluation of TRACED across $N$**
>
> We thank the reviewer for their careful reading of the paper and comments.
>
> ## 1. Alignment score improvements
>
> We agree with the reviewer that relying on BLIP alignment scores for optimization and evaluation introduces a risk of circularity. Unfortunately, standard caption-quality metrics such as CIDEr, METEOR, SPICE, BLEU, and ROUGE are not informative in our fine-grained error-correction setting: because noisy captions differ from the reference captions by only a few tokens, these overlap-based metrics remain artificially high even when the caption contains factual errors. Below, we report the extremely high, if not ideal scores obtained by these metrics on the initial noisy captions to illustrate this limitation.
>
> Test caption quality scores for the noisy captions in the dataset with 50\% fine-grained noise. Mean and standard error are computed over the same three seeds used in the main paper:
>
> | Metric | Flickr30k| MS COCO|
> |--------|-----------|---------|
> | BLEU-4 | 0.669 ± 0.005| 0.614 ± 0.002 |
> | ROUGE  | 0.808 ± 0.003| 0.792 ± 0.001 |
> | CIDEr  | 1.463 ± 0.012| 1.362 ± 0.004 |
> | METEOR | 0.449 ± 0.002| 0.420 ± 0.000 |
> | SPICE  | 0.257 ± 0.003| 0.249 ± 0.001 |
>
> Since the primary contribution of this work is to show that caption trajectories can be leveraged to improve error detection, we focus here only on showing how our interpretable tokens can be used to improve an alignment metric.
> However, we want to emphasize the fact that our correction pipeline is model-agnostic: it can optimize any alignment score. Therefore, we believe that our results are promising. As stronger or provably human-aligned metrics become available, the same procedure can be applied to these scoring functions.
> We agree that the correction quality could be improved by using a curated large-scale human evaluation dataset, which is beyond the scope of this paper.
>
> ## 2. Evaluating TRACED on real-world caption errors
>
> We use synthetic noise primarily for evaluation, as it allows us to construct labeled datasets where the impact of applying TRACED to an existing error-detection baseline can be measured easily. In contrast, evaluating the methods on real-world datasets is more challenging due to the lack of ground truth labels. However, we agree with the reviewer that relying exclusively on synthetic perturbations limits the strength of our empirical claims.
> Therefore, we conducted an additional experiment on real data. We applied both BLIP and TRACED-BLIP (calibrated on the datasets with 50\% fine-grained noise) to the original test sets of Flickr30k and MS COCO. Following the same approach as in LEMoN [1], we selected the 50 samples in each dataset that each method flagged as most erroneous and manually checked each of these samples. The results are reported below:
>
> Performance of TRACED-BLIP and BLIP on the original datasets. For each dataset and method, the top 50 samples most likely to contain errors (as predicted by the detection model) are manually labeled.
>
> | Dataset| TRACED-BLIP | BLIP |
> |-----------|--|------|
> | Flickr30k | 0.36| 0.32 |
> | MS COCO| 0.36| 0.30 |
>
> ## 3.1 Ablation on candidate count
>
> We agree with the reviewer that additional ablations would help clarify the contribution of each component of our approach. Therefore, we conducted additional ablation studies using Flickr30k to better isolate the effects of the key hyperparameters.
>
> Candidate count ($N$): We examined the role of N, the number of candidate edits explored at each step, for the GCD Algorithm. For the Elimination algorithm, $N$ is small and is equal by design to the number of tokens in the caption. $N$ decreases by one at each iteration, making exploration progressively faster.
>
> | Method  | Noise Type| N=32  | N=64  | N=128 | N=256 | N=512 |
> |--------|----|-------|-------|-------|-------|-------|
> | BLIP | fine-grained  | 0.751 | 0.759 | 0.762 | 0.755 | 0.769 |
> | BLIP | noun  | 0.943 | 0.941 | 0.943 | 0.942 | 0.941 |
> | BLIP | random| 0.981 | 0.979 | 0.978 | 0.979 | 0.980 |
> | TRACED-LEMoN_OPT| fine-grained  | 0.680 | 0.689 | 0.688 | 0.680 | 0.689 |
> | TRACED-LEMoN_OPT| noun  | 0.896 | 0.893 | 0.891 | 0.894 | 0.897 |
> | TRACED-LEMoN_OPT| random| 0.974 | 0.973 | 0.970 | 0.972 | 0.973 |
> | TRACED-CLIP  | fine-grained  | 0.679 | 0.681 | 0.668 | 0.686 | 0.690 |
> | TRACED-CLIP  | noun  | 0.885 | 0.894 | 0.893 | 0.894 | 0.893 |
> | TRACED-CLIP  | random| 0.970 | 0.971 | 0.970 | 0.968 | 0.971 |
>
> This table suggests that while larger values of $N$ can lead to better performance, particularly for more challenging noise types such as the fine-grained noise, smaller values of $N$ are often sufficient to achieve near-ideal performance. We will run this experiment with additional seeds to report standard errors, and we will include these results in the revised paper.

---

> ### Author Response · Authors · 2025-11-22
> **Evaluation of TRACED across T and c and discussion on exploration strategy**
>
> ## 3.2 Ablation on trajectory length
>
> Trajectory length ($T$): We evaluated the influence of trajectory length to determine how much information is gained from longer versus shorter paths. We used the Elimination algorithm to conduct this experiment.
> | Method| Noise Type| T=0| T=1| T=2| T=5| T=10  | T=15  | T=20  | T=30  | T=All |
> |-----|---|-|-|-|-|-|-|-|-|-|
> | TRACED-BLIP  | fine-grained | 74.2 ± 0.8 | 76.9 ± 0.3 | 76.3 ± 0.3 | 75.9 ± 0.1 | 76.5 ± 0.4 | 76.5 ± 0.1 | 76.6 ± 0.2 | 76.6 ± 0.3 | 76.6 ± 0.3 |
> | TRACED-BLIP  | noun | 93.6 ± 0.3 | 93.7 ± 0.4 | 93.7 ± 0.2 | 93.8 ± 0.4 | 93.8 ± 0.2 | 93.7 ± 0.3 | 93.8 ± 0.3 | 93.7 ± 0.3 | 93.8 ± 0.3 |
> | TRACED-BLIP  | random  | 97.8 ± 0.1 | 97.9 ± 0.3 | 97.8 ± 0.2 | 98.1 ± 0.1 | 98.1 ± 0.2 | 98.2 ± 0.0 | 98.2 ± 0.1 | 98.1 ± 0.1 | 98.2 ± 0.1 |
> | TRACED-LEMoN_OPT | fine-grained | 66.1 ± 0.7 | 67.8 ± 1.4 | 67.8 ± 1.8 | 68.2 ± 1.6 | 68.0 ± 1.2 | 68.4 ± 1.2 | 68.3 ± 1.4 | 68.6 ± 1.2 | 68.5 ± 1.0 |
> | TRACED-LEMoN_OPT | noun | 89.2 ± 0.3 | 89.7 ± 0.5 | 90.0 ± 0.4 | 89.9 ± 0.3 | 90.4 ± 0.1 | 90.6 ± 0.1 | 90.6 ± 0.2 | 90.6 ± 0.2 | 90.8 ± 0.1 |
> | TRACED-LEMoN_OPT | random  | 97.5 ± 0.1 | 97.4 ± 0.0 | 97.5 ± 0.1 | 97.5 ± 0.1 | 97.5 ± 0.0 | 97.6 ± 0.2 | 97.6 ± 0.1 | 97.5 ± 0.1 | 97.5 ± 0.1 |
> | TRACED-CLIP  | fine-grained | 64.8 ± 0.6 | 67.8 ± 0.7 | 68.6 ± 0.7 | 69.1 ± 0.5 | 69.4 ± 0.6 | 69.2 ± 0.6 | 68.9 ± 0.2 | 69.0 ± 0.4 | 68.9 ± 0.4 |
> | TRACED-CLIP  | noun | 89.3 ± 0.3 | 89.9 ± 0.4 | 89.9 ± 0.3 | 89.9 ± 0.2 | 90.1 ± 0.1 | 91.1 ± 0.3 | 90.9 ± 0.2 | 90.7 ± 0.3 | 90.7 ± 0.2 |
> | TRACED-CLIP  | random  | 97.2 ± 0.1 | 97.3 ± 0.2 | 97.4 ± 0.1 | 97.4 ± 0.2 | 97.5 ± 0.1 | 97.6 ± 0.1 | 97.5 ± 0.0 | 97.6 ± 0.1 | 97.6 ± 0.1 |
>
> This table suggests that removing only a few tokens, or even just one for BLIP, is enough to reach optimal performance. This is a strong result for our method. Although running the procedure on the full sentence is needed for interpretability at the token-level, the process can be greatly shortened when the goal is limited to error detection.
>
> ## 3.3 Ablation on semantic similarity score
>
> Choice of semantic similarity score ($c$): We also studied the impact of the semantic similarity function $c$. For BLIP, we compared using the ITC component (as in the main paper) with using CLIP instead. For both LEMoN and CLIP, we performed the other substitution and evaluated whether replacing CLIP with BLIP (ITC) affected performance. The resulting comparisons are provided below.
>
> | s| Noise Type| c = CLIP | c = BLIP (ITC) |
> |---|---|-----|-------|
> | BLIP (ITM)| fine-grained | 76.0 ± 0.2| 76.6 ± 0.3  |
> | BLIP (ITM)| noun | 93.7 ± 0.3| 93.8 ± 0.3  |
> | BLIP (ITM)| random  | 98.0 ± 0.1| 98.2 ± 0.1  |
> | LEMON_OPT| fine-grained | 68.5 ± 1.0| 69.2 ± 1.0  |
> | LEMON_OPT| noun | 90.8 ± 0.1| 90.5 ± 0.2  |
> | LEMON_OPT| random  | 97.5 ± 0.1| 97.4 ± 0.3  |
> | CLIP | fine-grained | 68.9 ± 0.4| 70.0 ± 0.2  |
> | CLIP | noun | 90.7 ± 0.2| 90.9 ± 0.3  |
> | CLIP | random  | 97.6 ± 0.1| 97.5 ± 0.3  |
>
> While the choice of BLIP (ITC) seems preferable for $s=$BLIP (ITM), the choice of $c$ does not seem to have a decisive impact on the final performance, and both CLIP and BLIP (ITC) appear to be effective metrics $c$ for error-detection.
>
> We will include all these results in the revised paper and thank the reviewer for highlighting the need for these ablations.
>
> Regarding classifier architecture, we agree that a more systematic search could yield further improvements. However, the central goal of our work is to demonstrate the value of trajectory information, and we believe the strong performance of XGBoost already supports our main claim: caption trajectories provide a substantially richer signal than a single alignment score and significantly improve error detection.
>
> ## 4. Discussion on the exploration strategies
>
> We agree that a more detailed discussion of the different exploration strategies would strengthen the paper. We reported results for all three algorithms (Elimination, GCD, and Fast GCD) because their comparison is informative to the reader: token elimination and token replacement are both natural ways of exploring the caption space. However, our experiments on BLIP, CLIP, and LEMoN indicate that the three methods extract trajectory information of similar utility: substituting tokens (as in GCD/Fast GCD) and removing them (as in Elimination) both allow us to identify erroneous captions effectively. Consequently, Elimination emerges as the most practical choice, offering comparable detection performance at lower computational cost.

---

### Official Review · Reviewer_ZqLU · 2025-10-31

**Soundness:** 3
**Presentation:** 2
**Contribution:** 3
**Rating:** 4
**Confidence:** 4

**Summary:**

This paper aims to detect and correct errors in vision-text data pairs, which is an interesting issue. To tackle this issue, a trajectory-guided method, named TRACED, is proposed, which edits the caption iteratively. This is a plug-and-play framework for different scoring multi-modality models. Experiments show that such a detection and correction manner is helpful and useful.

**Strengths:**

1.	The motivation of this paper is interesting.
2.	The proposed TRACED model is effective in detecting caption errors and finding the most influential tokens in a sentence.
3.	The experimental results demonstrate the effectiveness of this model.

**Weaknesses:**

1.	From my perspective, it’s true that current vision-text training data may contain mistakes, which affects the model's learning. But the authors don’t provide the number and the proportion of mislabelled instances of a specific dataset, like MS-COCO, Flickr30K. Hence, I have doubts about whether the pre-trained datasets have this issue.
2.	Table 1 is not easy to read and follow.
3.	In Appendix A.4, the authors propose to construct a new benchmark, but I cannot see more details about this benchmark.
4.	This paper is able to detect errors in captions. Hence, I think it would be better to represent how the correction procedure improves the multi-modal models’ understanding performance, such as the image-text retrieval task, the captioning task, etc. For now, Figure 12 still evaluates the data itself with different models after correction.

**Questions:**

1.	I cannot fully understand Table 1. Take the Flickr30K dataset as an example. What does the AUC here mean? Then what about the improvement? What’s the baseline result to compute the improvement?
2.	For each image-text data pair, how many noised captions does the pipeline generate?


------
This motivation is interesting to me. I will be happy to raise my score if my concerns can be solved.

---

> ### Author Response · Authors · 2025-11-22
> **Dataset quality and evaluation protocol**
>
> We thank the reviewer for their careful reading of the paper and comments.
>
> ## 1. Prevalence of annotation errors
> As the reviewer noted, existing image–caption datasets can contain mistakes. This issue is particularly pronounced in large-scale datasets created via web scraping [1] or synthetic generation [2].
> To study error detection in a controlled setting, we follow the approach of [3]. We begin with high-quality datasets such as Flickr30k and MS COCO, both produced by human annotators, and we treat them as error-free. We then introduce controlled synthetic noise to construct labeled datasets for evaluating the different error-detection methods.
>
> However, we agree with the reviewer that these datasets are not necessarily immune to errors. To estimate the prevalence of such annotation errors, we randomly sampled 100 image–caption pairs from each dataset and observed 2 errors in Flickr30k and 6 in MS COCO.
> Although the proportion appears to be small, developing ways to account for them would be an interesting direction for future work (as also noted in [3])
>
> ## 2. Evaluation protocol in Table 1
>
> After adding synthetic noise (either our new fine-grained noise, or classic caption permutations as in [3], we obtain a labeled dataset in which we know which captions are original (correct) and which are modified (incorrect). We then evaluate the error-detection baselines (CLIP, LEMoN, BLIP) on this dataset, and our framework which builds on top of each baseline (TRACED-CLIP, TRACED-LEMoN, TRACED-BLIP). Because our framework supports multiple exploration strategies, we report performance for each of them.
> The accuracy (edited on nov 26th) values reflect error-detection performance on this binary task (distinguishing erroneous captions from the correct ones). The “Improvement” column shows the relative accuracy (edited on nov 26th) gain of TRACED over the corresponding baseline (e.g., TRACED-BLIP over BLIP). We will provide some additional details in the paper to make it clearer.
>
> [1] Radford, A. et al. (2021). Learning transferable visual models from natural language supervision. In ICML 2021.
>
> [2] Li, J. et al. (2022). BLIP: Bootstrapping language–image pre-training for unified vision–language understanding and generation. In ICML 2022.
>
> [3] Zhang, H. et al. (2025). LEMoN: Label Error Detection using Multimodal Neighbors. In ICML 2025.

---

> ### Author Response · Authors · 2025-11-22
> **Clarifications on noise generation and impact of filtering and correction on downstream performance**
>
> ## 3. Clarifications on fine-grained noise generation
>
> Thank you for pointing this out. Based on your second question, we believe you are referring to the lack of clarity regarding the number of noisy captions generated with fine-grained noise. In our pipeline, we generate 20 candidate sentences (K=20) and select the two worst according to the alignment score. We will clarify these hyperparameters in Section A.4.
> We believe the other steps of the generation process are described in the paper: we ask gpt-o4-mini to produce 20 erroneous captions per original caption using the prompt in Section A.14, then we filter these outputs to retain the most erroneous captions according to AlignScore, a Natural Language Inference model. Please let us know if any additional details would be helpful.
>
> ## 4. Impact of filtering and correction on downstream captioning performance
>
> We agree with the reviewer that examining the impact of error detection and correction on downstream performance can be interesting to the reader. To investigate this, we fine-tuned BLIP-2 [4] on (i) the original datasets containing 50\% fine-grained noise and (ii) cleaned variants obtained by removing sentences flagged as incorrect by BLIP (the strongest baseline) or by TRACED-BLIP. In addition to filtering, we also tested the correction strategy, where sentences predicted as wrong are replaced with corrected versions produced either by InternVL3-1B alone or using the token-level correction procedure described in Section 3.4.
>
> Following [3], we fine-tuned the model using LoRA (rank = 4) and report BLIP, BLEU-4, ROUGE, and CIDEr scores. We trained the models for 3 epochs, with a learning rate of 1e-5, AdamW and using early stopping. The table below summarizes the results:
>
> | Cleaning Method  | BLIP  | BLEU-4 | ROUGE | CIDEr |
> |------------|-------|--------|-------|-------|
> | Noisy Dataset| 0.656 | 0.312  | 0.563 | 0.687 |
> | Filtering – BLIP | 0.675 | 0.303  | 0.559 | 0.662 |
> | Filtering – TRACED-BLIP  | 0.687 | 0.317  | 0.564 | 0.692 |
> | Ideal Filtering  | 0.678 | 0.317  | 0.566 | 0.694 |
> | Correction InternVL3-1B  | 0.674 | 0.321  | 0.568 | 0.708 |
> | Correction TRACED-InternVL3-1B| 0.680 | 0.324  | 0.569 | 0.706 |
>
> This table shows the usefulness of our procedure.
>
> Filtering with TRACED-BLIP achieves performance very close to Ideal Filtering and outperforms the baseline (BLIP-filtering) across all metrics. Applying caption correction provides further improvements in BLEU-4, ROUGE, and CIDEr, indicating higher-quality captions, and using TRACED brings additional gains in BLIP score. This suggests that the corrected captions are not only closer to the ground-truth references but also better aligned with the images and contain fewer errors according to BLIP. We will also run additional seeds to report standard errors in the paper.
>
> ## Questions:
>
> Thank you for taking the time to write these additional questions. We believe we have addressed them above, but please let us know if you have any other questions you would like us to answer.
>
> [4] Li, J. et al. (2023). BLIP-2: Bootstrapping language–image pre-training with frozen image encoders and large language models. In ICML 2023.

---

> > ### Comment · Reviewer_ZqLU · 2025-11-24
> >
> > After reading the authors' rebuttal, my concerns have been addressed. Hence, I recommend acceptance.
> >
> > Besides, I suggest redrawing Table 1 for better reading.

---

### Official Review · Reviewer_QGRZ · 2025-10-31

**Soundness:** 3
**Presentation:** 2
**Contribution:** 3
**Rating:** 4
**Confidence:** 4

**Summary:**

This paper proposes TRACED (Trajectory Creation for Error Detection), a framework for detecting errors in image-caption datasets. Unlike existing methods that rely on a single similarity score, TRACED iteratively edits captions to create a "trajectory" and analyzes how alignment scores and semantic similarity evolve. The key insight is that correct captions stabilize after minor edits, while erroneous captions show substantial improvement potential.

**Main contributions:**
1. A novel trajectory-based approach using token deletion (Elimination) and replacement (GCD, Fast GCD)
2. A model-agnostic framework applicable to existing methods (CLIP, LEMoN, BLIP)
3. Introduction of fine-grained noise generated by GPT-4o-mini, more realistic than random/noun swaps
4. Up to 2.8% AUC improvement on MS COCO, Flickr30k, and MM-IMDb
5. Token-level error localization for interpretability and caption correction, achieving up to 14.5% alignment score improvement

**Strengths:**

1. **Novel and intuitive approach**: The trajectory-based idea is creative and well-motivated. The insight that correct captions require minimal edits while erroneous ones show substantial improvement is clear and compelling. The framework is model-agnostic and consistently improves multiple baselines (CLIP, LEMoN, BLIP).

2. **Practical value with interpretability**: Beyond detection, TRACED provides token-level error localization and enables caption correction. The 14.5% alignment score improvement on InternVL3-14B demonstrates practical utility.

3. **More realistic evaluation**: The fine-grained noise benchmark better reflects real annotation errors than wholesale caption swaps. The finding that TRACED's largest gains occur under fine-grained noise (where baselines struggle most) validates the approach's practical relevance.

**Weaknesses:**

1. **Limited demonstration of generality (Critical)**: The same "no jacket" example appears repeatedly (Figures 1, 3, and text), raising cherry-picking concerns. The paper lacks diverse examples showing different error types (object misidentification, color errors, numerical errors, relationship errors, distributed errors) and failure cases. This is essential for convincing readers of the method's generality, especially given concerns about ambiguous images, subjective captions, and varying caption lengths.

2. **Circular reasoning and evaluation concerns**: Caption correction is evaluated using BLIP alignment scores while being optimized with BLIP-based TRACED. Independent metrics (METEOR, CIDEr, SPICE) or human evaluation are needed to verify whether "correction" actually fixes factual errors rather than simply adjusting to BLIP's preferences. Additionally, critical results are relegated to the appendix: noise-type breakdown (Table 4), trajectory importance (A.12), and maximization vs. minimization (A.13) should be in the main text to support key claims.

3. **Insufficient validation of fine-grained noise**: While conceptually appealing, the fine-grained noise lacks validation against real annotation errors. The use of GPT-4o-mini raises reproducibility concerns, and the omission of MM-IMDb due to cost limits generalizability claims. The 50% noise rate is also unrealistic, performance at lower rates (10%, 20%) is needed. Furthermore, key questions remain unaddressed: Why is maximization better than minimization? What about token addition (not just deletion/replacement)? What is the sensitivity to hyperparameters N and T?

**Questions:**

1. Can you provide diverse examples showing different error types (object, color, number, relationship, distributed errors) and failure cases beyond the repeated "no jacket" example?

2. How did you validate that GPT-4o-mini generated noise reflects real annotation errors? Can you compare against datasets with actual annotation errors (e.g., Conceptual Captions)?

3. Can you evaluate caption correction using independent metrics (CIDEr, SPICE) or human evaluation, not just BLIP scores? Does "correction" fix factual errors or just adjust to BLIP preferences?

4. Why are critical ablation results (Table 4 on noise types, A.12 on trajectory importance, A.13 on maximization vs. minimization) in the appendix rather than the main text? Can you provide specific numbers showing fine-grained noise improvements in the main paper?

5. How does the method perform at realistic noise rates (10%, 20%) rather than 50%? What about very long/short captions, ambiguous images, or subjective descriptions?

6. Why not consider token addition, only deletion/replacement? What is the sensitivity to N and T? Why does maximization outperform minimization (A.13 only says "seem to be obtained")?

---

> ### Author Response · Authors · 2025-11-22
> **Diverse examples can be found in the new benchmark**
>
> We thank the reviewer for their careful reading of the paper and comments.
>
> ## 1.1 Fine-grained errors contain diverse errors
>
> We used the same “no jacket” example in multiple figures and discussions to allow the reader to directly compare how different exploration strategies behave on an identical input, and to clearly contrast our method with the baseline. However, we agree that a broader set of examples would be beneficial to the paper and would allow us to show the full range of error types, beyond negations.
> To address this, we examined the generated samples with fine-grained noise in Flickr30k and collected examples per error type (wrong object, action, color, number, negations, and distributed errors). Below is the set of examples we gathered, which we will add to the Appendix. We will also include the corresponding images and provide a similar list for MS COCO.
>
> ### Wrong object / element
>
> | True sentence | Noisy sentence |
> |----|-----|
> | A black and white dog is running through the grass. | A black and white cat is running through the grass. |
> | Several students waiting outside an igloo. | Several students waiting outside a car. |
> | A man in a white shirt stands high up on scaffolding. | A woman in a red shirt stands high up on scaffolding. |
>
> ### Wrong number
>
> | True sentence | Noisy sentence |
> |----|-----|
> | A group of spectators watch a men's sand volleyball game. | A solitary spectator watches a men's sand volleyball game. |
> | Shaft of light in a cave shows three spelunkers. | A shaft of light in a cave reveals one spelunker. |
> | Person standing on rocky edge of water with hilly land in background. | A group of people standing on the sandy beach with flat land in the background. |
>
> ### Wrong color
>
> | True sentence | Noisy sentence |
> |----|-----|
> | A boy wearing a orange Doritos jersey jumps up in the air. | A girl wearing a blue Doritos jersey jumps up in the air. |
> | A brown dog about to catch a green Frisbee. | A black dog about to catch a red Frisbee. |
> | A man in brown building a raft. | A man in red building a raft. |
>
> ### Wrong action
>
> | True sentence | Noisy sentence |
> |----|-----|
> | Girl wearing blue shirt and black shorts plays trumped outside. | The girl wearing a blue shirt and black shorts sings outside. |
> | One man holds another man's head down and prepares to punch him in the face. | One man holds another man's head down while his friends cheer him on. |
> | A group of people are hiking up an icy hillside. | A group of people are skiing down an icy hillside. |
>
> ### Wrong negation
>
> | True sentence | Noisy sentence |
> |----|-----|
> | Three people are sitting at an outside picnic bench with an umbrella. | Two people are sitting at an inside picnic table without an umbrella. |
> | A man is standing in front of a brick storefront wearing a black jacket. | A man is standing in front of a brick storefront wearing no jacket. |
> | A man playing an acoustic guitar and singing with a group of people behind him including a woman who is singing along. | A man not playing any instrument and speaking with a group of people behind him including a woman who is taking notes. |
>
> ### Distributed errors
>
> | True sentence | Noisy sentence |
> |----|-----|
> | A bride in a light pink dress poses for a picture with male relatives and is being photographed by a man in a cream shirt with white pants. | A bride in a light pink dress poses for a picture with her siblings and is being photographed by a girl in a green shirt with a floral skirt. |
> | A band is of four members including a woman and three men are playing their instruments with an open guitar case in front of them. | An orchestra is of ten members including a woman and nine men are playing their instruments with an open violin case in front of them. |
> | A young man in a gray tee-shirt and gray sweatpants stands by a metal tiered shelf in an industrial kitchen, holding the top edge of the metal structure, with one leg resting on the knee of the other leg. | A young woman in a blue tee-shirt and black sweatpants stands by a metal tiered shelf in an industrial kitchen, holding the bottom edge of the metal structure, with one leg resting on the knee of the other leg. |

---

> > ### Author Response · Authors · 2025-11-22
> > **Similar caption lengths are observed for the original and fine-grained noise. Classic caption quality metrics are artificially high.**
> >
> > ## 1.2 Caption Length statistics for Flickr30k and MS COCO:
> >
> > Regarding caption length, because we instruct the LLM to introduce errors while making only minimal changes to the reference caption, the resulting captions tend to have nearly the same number of tokens as the originals. We also report below summary statistics on caption lengths for both the original captions and the synthetically generated fine-grained noisy captions.
> >
> > | Metric| Flickr30k Correct | Flickr30k Noisy | MS COCO Correct | MS COCO Noisy |
> > |--------|--------|-------|-------|-----|
> > | Mean length  | 12.25| 12.30| 10.45| 10.50  |
> > | Std. dev. | 5.15 | 4.99| 2.37| 2.43|
> > | Min length| 2.00 | 2.00| 7.00| 5.00|
> > | 25th percentile| 9.00 | 9.00| 9.00| 9.00|
> > | Median (50th)| 11.00| 11.00| 10.00| 10.00  |
> > | 75th percentile| 15.00| 15.00| 11.00| 12.00  |
> > | Max length| 69.00| 66.00| 47.00| 48.00  |
> >
> > ## 2 Standard caption quality metrics are artificially high
> >
> > For the caption-correction stage, we focus on fine-grained noise because we believe correction is only meaningful when captions contain subtle and more realistic errors; random caption swaps typically produce captions that are entirely incorrect, turning the task into image captioning from scratch rather than correction.
> >
> > We agree that independent caption-quality metrics would strengthen the evaluation. However, fine-grained noise makes this challenging: since the noisy captions differ from the reference captions by only a few tokens, metrics such as METEOR, CIDEr, and SPICE—which largely reward token overlap—remain artificially high even when the caption contains a factual error.
> >
> > We include below the scores obtained for these metrics on the initial noisy captions.
> > These scores are extremely high, if not ideal, despite the errors introduced. Therefore, these metrics won't allow us to properly measure the impact of our corrections.
> >
> > Test caption quality scores for the noisy captions in the dataset with 50\% fine-grained noise. Mean and standard error are computed over the same three seeds used in the main paper:
> >
> > | Metric | Flickr30k| MS COCO|
> > |--------|-----------|---------|
> > | BLEU-4 | 0.669 ± 0.005| 0.614 ± 0.002 |
> > | ROUGE  | 0.808 ± 0.003| 0.792 ± 0.001 |
> > | CIDEr  | 1.463 ± 0.012| 1.362 ± 0.004 |
> > | METEOR | 0.449 ± 0.002| 0.420 ± 0.000 |
> > | SPICE  | 0.257 ± 0.003| 0.249 ± 0.001 |
> >
> > More broadly, we acknowledge the reviewer’s concern about using a BLIP-based score both for optimization and evaluation.
> > Our primary goal and claim in the paper is to show how caption trajectories can be used to improve error detection and provide interpretable results.
> > The correction pipeline is included to illustrate how our framework can be used to boost an alignment metric.
> >
> > Importantly, our framework is model-agnostic: it can optimize any alignment score. As stronger or more human-aligned metrics become available, they can be substituted into our pipeline.
> >
> > We agree the correction quality could be improved by using a curated large-scale human evaluation dataset, which is beyond the scope of this paper.
> >
> > Finally, we agree that several key ablations, which provide essential support for our claims, have been placed in the Appendix due to space constraints. We will move them into the main text in the revised version.

---

> ### Author Response · Authors · 2025-11-22
> **Quality of the fine-grained noise and ablation studies**
>
> # Message 3: Quality of the fine-grained noise and ablation studies
>
> ## 3.1 Quality of the fine-grained noise and reproducibility
>
> Given the size of the dataset, a full manual evaluation would require substantial effort that is beyond the scope of this work. Instead, we provide representative examples that illustrate the range and quality of the generated errors. Although not exhaustive, our examination suggests that the dataset contains numerous subtle and challenging errors pertinent to our task.
>
> While we understand the reproducibility concern due to the use of a proprietary model, we have provided the prompts used and we will make sure to open source the dataset that we generated upon paper acceptance.
> ## 3.2 Evaluation of TRACED across noise types
>
> We focused on 50\% in the main text in order to showcase the performance of our framework in the case of a balanced dataset. However, we agree that a broader evaluation across noise levels, including lower ones, is valuable for the reader. The results for TRACED (using the Elimination Algorithm) across noise ratios $\alpha$ on Flickr30k are presented below.
>
> As we noticed an oversight in the AUC/Accuracy, we provide below both the Test AUC and Accuracy across noise levels (edited on nov 26th).
>
> Mean AUC (in percent) along with the standard error over three random seeds.
>
> | Noise type | Method| 0.1| 0.2| 0.3| 0.4|
> |--|--|---|---|---|---|
> | Fine-grained | Blip| 80.8 ± 1.0| 79.1 ± 0.6| 80.6 ± 1.1| 81.4 ± 0.7|
> | Fine-grained | Traced Blip| 83.5 ± 1.5| 82.6 ± 0.8| 84.0 ± 1.3| 84.4 ± 0.8|
> | Fine-grained | Lemon Opt| 73.5 ± 2.1| 70.7 ± 0.9| 71.4 ± 0.8| 72.1 ± 1.5|
> | Fine-grained | Traced Lemon Opt| 76.4 ± 1.1| 75.4 ± 2.1| 75.0 ± 0.7| 74.7 ± 1.0|
> | Fine-grained | Clip| 71.5 ± 2.5| 70.0 ± 1.5| 70.7 ± 1.3| 70.6 ± 1.2|
> | Fine-grained | Traced Clip| 76.7 ± 1.4| 76.0 ± 1.8| 75.4 ± 1.0| 75.8 ± 1.1|
> | Noun| Blip| 99.0 ± 0.3| 98.8 ± 0.2| 98.8 ± 0.2| 98.5 ± 0.2|
> | Noun| Traced Blip| 99.1 ± 0.3| 98.8 ± 0.2| 98.8 ± 0.2| 98.5 ± 0.2|
> | Noun| Lemon Opt| 95.7 ± 0.5| 95.6 ± 0.8| 95.3 ± 0.4| 94.5 ± 0.1|
> | Noun| Traced Lemon Opt| 96.2 ± 0.5| 95.9 ± 0.7| 96.0 ± 0.3| 95.1 ± 0.2|
> | Noun| Clip| 95.5 ± 0.3| 95.6 ± 0.7| 95.4 ± 0.3| 94.7 ± 0.3|
> | Noun| Traced Clip| 96.4 ± 0.3| 96.3 ± 0.4| 96.2 ± 0.3| 95.5 ± 0.2|
> | Random| Blip| 99.6 ± 0.3| 99.7 ± 0.1| 99.7 ± 0.1| 99.7 ± 0.1|
> | Random| Traced Blip| 99.7 ± 0.2| 99.6 ± 0.1| 99.8 ± 0.1| 99.8 ± 0.1|
> | Random| Lemon Opt| 99.4 ± 0.1| 99.5 ± 0.2| 99.3 ± 0.3| 99.4 ± 0.1|
> | Random| Traced Lemon Opt| 99.5 ± 0.1| 99.6 ± 0.2| 99.5 ± 0.1| 99.5 ± 0.1|
> | Random| Clip| 99.3 ± 0.2| 99.5 ± 0.2| 99.5 ± 0.1| 99.5 ± 0.1|
> | Random| Traced Clip| 99.3 ± 0.2| 99.6 ± 0.1| 99.6 ± 2.0| 99.6 ± 0.1|
>
> Mean Accuracy (in percent) along with the standard error over three random seeds.
> | Noise type | Method| 0.1| 0.2| 0.3| 0.4|
> |--|--|---|---|---|---|
> | Fine-grained | Blip| 91.4 ± 0.3| 83.1 ± 0.2| 79.1 ± 0.5| 77.2 ± 0.8|
> | Fine-grained | Traced Blip| 91.6 ± 0.3| 85.1 ± 0.8| 81.1 ± 1.0| 78.1 ± 0.7|
> | Fine-grained | Lemon Opt| 90.1 ± 0.1| 80.8 ± 0.1| 73.3 ± 0.3| 68.6 ± 0.9|
> | Fine-grained | Traced Lemon Opt| 90.5 ± 0.2| 82.1 ± 0.3| 75.0 ± 0.4| 70.2 ± 0.9|
> | Fine-grained | Clip| 90.0 ± 0.0| 81.2 ± 0.3| 73.3 ± 0.3| 68.6 ± 0.9|
> | Fine-grained | Traced Clip| 90.3 ± 0.3| 82.2 ± 0.2| 75.7 ± 0.2| 70.6 ± 0.8|
> | Noun| Blip| 97.7 ± 0.1| 95.9 ± 0.4| 95.0 ± 0.1| 94.2 ± 0.3|
> | Noun| Traced Blip| 97.7 ± 0.2| 96.0 ± 0.2| 95.0 ± 0.2| 94.5 ± 0.2|
> | Noun| Lemon Opt| 96.3 ± 0.2| 94.1 ± 0.5| 91.9 ± 0.5| 89.5 ± 0.1|
> | Noun| Traced Lemon Opt| 96.4 ± 0.2| 94.2 ± 0.4| 92.6 ± 0.2| 90.6 ± 0.2|
> | Noun| Clip| 96.0 ± 0.2| 94.0 ± 0.4| 91.7 ± 0.2| 89.8 ± 0.4|
> | Noun| Traced Clip| 96.4 ± 0.3| 94.3 ± 0.4| 92.5 ± 0.2| 90.9 ± 0.1|
> | Random| Blip| 99.0 ± 0.3| 98.4 ± 0.1| 98.1 ± 0.2| 97.7 ± 0.3|
> | Random| Traced Blip| 98.8 ± 0.3| 98.3 ± 0.2| 97.9 ± 0.1| 97.8 ± 0.4|
> | Random| Lemon Opt| 98.3 ± 0.2| 97.7 ± 0.4| 97.2 ± 0.2| 96.8 ± 0.4|
> | Random| Traced Lemon Opt| 98.5 ± 0.2| 97.9 ± 0.4| 97.8 ± 0.1| 97.1 ± 0.3|
> | Random| Clip| 98.1 ± 0.2| 97.8 ± 0.4| 97.2 ± 0.1| 96.7 ± 0.3|
> | Random| Traced Clip  | 98.6 ± 0.2| 97.7 ± 0.4| 97.8 ± 0.1| 96.9 ± 0.3|
>
> We observe that TRACED consistently improves over the baselines across all noise ratios.
>
> ## 3.3 Maximization or Minimization
>
> Empirically we find that maximization tends to yield stronger results. We believe this can be explained by the fact that maximization produces more informative trajectories: removing incorrect tokens can momentarily fix an incorrect caption, before too many tokens are removed, leading to an incorrect → correct → incorrect pattern. In contrast, minimization will likely remove correct tokens first, making the caption incorrect very early. Subsequent removal of the erroneous tokens then produces little additional signal, as the caption is already incorrect.

---

> ### Author Response · Authors · 2025-11-22
> **Evaluation of TRACED across N and T and discussion on exploration strategy**
>
> ## 3.4 Evaluation of TRACED across $N$ and $T$
>
> We also agree with the reviewer that additional ablation studies would be useful for T and N. We examined the role of N, the number of candidate edits explored at each step, for the GCD Algorithm. For the Elimination algorithm, $N$ is small and is equal by design to the number of tokens in the caption. $N$ decreases by one at each iteration, making exploration progressively faster.
>
> We will also run additional seeds to provide standard errors.
>
> | Method  | Noise Type| N=32  | N=64  | N=128 | N=256 | N=512 |
> |--------|----|-------|-------|-------|-------|-------|
> | BLIP | fine-grained  | 0.751 | 0.759 | 0.762 | 0.755 | 0.769 |
> | BLIP | noun  | 0.943 | 0.941 | 0.943 | 0.942 | 0.941 |
> | BLIP | random| 0.981 | 0.979 | 0.978 | 0.979 | 0.980 |
> | TRACED-LEMoN_OPT| fine-grained  | 0.680 | 0.689 | 0.688 | 0.680 | 0.689 |
> | TRACED-LEMoN_OPT| noun  | 0.896 | 0.893 | 0.891 | 0.894 | 0.897 |
> | TRACED-LEMoN_OPT| random| 0.974 | 0.973 | 0.970 | 0.972 | 0.973 |
> | TRACED-CLIP  | fine-grained  | 0.679 | 0.681 | 0.668 | 0.686 | 0.690 |
> | TRACED-CLIP  | noun  | 0.885 | 0.894 | 0.893 | 0.894 | 0.893 |
> | TRACED-CLIP  | random| 0.970 | 0.971 | 0.970 | 0.968 | 0.971 |
>
> This table suggests that while larger values of $N$ can lead to better performance, particularly for more challenging noise types such as the fine-grained noise, smaller values of $N$ are often sufficient to achieve near-ideal performance.
>
> We perform a similar ablation study for $T$ using the Elimination algorithm on Flickr30k.
> | Method| Noise Type| T=0| T=1| T=2| T=5| T=10  | T=15  | T=20  | T=30  | T=All |
> |-----|---|-|-|-|-|-|-|-|-|-|
> | TRACED-BLIP  | fine-grained | 74.2 ± 0.8 | 76.9 ± 0.3 | 76.3 ± 0.3 | 75.9 ± 0.1 | 76.5 ± 0.4 | 76.5 ± 0.1 | 76.6 ± 0.2 | 76.6 ± 0.3 | 76.6 ± 0.3 |
> | TRACED-BLIP  | noun | 93.6 ± 0.3 | 93.7 ± 0.4 | 93.7 ± 0.2 | 93.8 ± 0.4 | 93.8 ± 0.2 | 93.7 ± 0.3 | 93.8 ± 0.3 | 93.7 ± 0.3 | 93.8 ± 0.3 |
> | TRACED-BLIP  | random  | 97.8 ± 0.1 | 97.9 ± 0.3 | 97.8 ± 0.2 | 98.1 ± 0.1 | 98.1 ± 0.2 | 98.2 ± 0.0 | 98.2 ± 0.1 | 98.1 ± 0.1 | 98.2 ± 0.1 |
> | TRACED-LEMoN_OPT | fine-grained | 66.1 ± 0.7 | 67.8 ± 1.4 | 67.8 ± 1.8 | 68.2 ± 1.6 | 68.0 ± 1.2 | 68.4 ± 1.2 | 68.3 ± 1.4 | 68.6 ± 1.2 | 68.5 ± 1.0 |
> | TRACED-LEMoN_OPT | noun | 89.2 ± 0.3 | 89.7 ± 0.5 | 90.0 ± 0.4 | 89.9 ± 0.3 | 90.4 ± 0.1 | 90.6 ± 0.1 | 90.6 ± 0.2 | 90.6 ± 0.2 | 90.8 ± 0.1 |
> | TRACED-LEMoN_OPT | random  | 97.5 ± 0.1 | 97.4 ± 0.0 | 97.5 ± 0.1 | 97.5 ± 0.1 | 97.5 ± 0.0 | 97.6 ± 0.2 | 97.6 ± 0.1 | 97.5 ± 0.1 | 97.5 ± 0.1 |
> | TRACED-CLIP  | fine-grained | 64.8 ± 0.6 | 67.8 ± 0.7 | 68.6 ± 0.7 | 69.1 ± 0.5 | 69.4 ± 0.6 | 69.2 ± 0.6 | 68.9 ± 0.2 | 69.0 ± 0.4 | 68.9 ± 0.4 |
> | TRACED-CLIP  | noun | 89.3 ± 0.3 | 89.9 ± 0.4 | 89.9 ± 0.3 | 89.9 ± 0.2 | 90.1 ± 0.1 | 91.1 ± 0.3 | 90.9 ± 0.2 | 90.7 ± 0.3 | 90.7 ± 0.2 |
> | TRACED-CLIP  | random  | 97.2 ± 0.1 | 97.3 ± 0.2 | 97.4 ± 0.1 | 97.4 ± 0.2 | 97.5 ± 0.1 | 97.6 ± 0.1 | 97.5 ± 0.0 | 97.6 ± 0.1 | 97.6 ± 0.1 |
>
> This table suggests that removing only a few tokens, or even just one for BLIP, is enough to reach optimal performance.
> This demonstrates a key advantage of our procedure in terms of reduced runtime.
>
> ## 3.5 Exploration strategy with eliminations and replacements
>
> Regarding the exploration strategies, our goal was to show how trajectory-based exploration can improve error detection. We focused on eliminations and replacements because they are sufficient to correct captions containing erroneous tokens. However, we agree that additional strategies, such as token addition, could also be valuable to explore in future work.

---

> ### Author Response · Authors · 2025-11-22
> **TRACED remains competitive across caption lengths; fine-grained noise results will be added to the main paper**
>
> ## Questions:
>
> We want to thank the reviewer for taking the time to provide these additional questions. We believe we have addressed most of them in the first part of our response, and we address below the remaining points:
>
> 4. We agree that the fine-grained noise results should be included in the main paper. These are our strongest results and will better highlight the usefulness of the method.
>
> 5. We agree that assessing the method under more different conditions is important. While evaluating TRACED on ambiguous images or subjective captions is difficult, we do examine its performance across caption lengths and compare it with the baselines; the results are shown below. We will include these results in the Appendix, as they indicate that TRACED performs consistently better across caption lengths.
>
> Test Accuracy (edited on nov 26th) per caption length for MS COCO. The bins correspond, in order, to the (0\%, 25\%], (25\%, 50\%], (50\%, 75\%], (75\%, 95\%] and (95\%, 100\%] percentiles
>
> | Noise type| Method  | (6.999, 9.0]| (9.0, 10.0] | (10.0, 11.0]| (11.0, 14.0]| (14.0, 34.0]|
> |----|------------|--------|--------|--------|--------|--------|
> | fine-grained  | BLIP  | 76.5 ± 0.3| 78.3 ± 0.9| 76.8 ± 0.7| 77.8 ± 0.4| 74.4 ± 2.0|
> | fine-grained  | TRACED - BLIP | 80.5 ± 0.5| 80.9 ± 1.2| 80.1 ± 0.5| 80.2 ± 0.2| 79.1 ± 0.4|
> | fine-grained  | LEMON_OPT| 68.6 ± 0.7| 68.5 ± 1.0| 68.0 ± 1.3| 69.6 ± 1.0| 68.1 ± 1.1|
> | fine-grained  | TRACED - LEMON_OPT| 70.5 ± 0.9| 70.4 ± 0.9| 71.1 ± 0.5| 71.2 ± 0.5| 71.3 ± 1.5|
> | fine-grained  | CLIP  | 66.7 ± 0.4| 66.8 ± 0.9| 65.0 ± 0.8| 68.1 ± 0.3| 65.4 ± 2.0|
> | fine-grained  | TRACED - CLIP | 70.3 ± 0.5| 68.9 ± 0.3| 69.8 ± 1.2| 70.3 ± 0.6| 68.5 ± 1.3|
> | noun  | BLIP  | 91.7 ± 0.4| 91.8 ± 0.1| 92.1 ± 0.2| 90.9 ± 0.4| 92.1 ± 0.9|
> | noun  | TRACED - BLIP | 91.9 ± 0.3| 92.6 ± 0.1| 92.6 ± 0.1| 91.6 ± 0.3| 91.7 ± 0.6|
> | noun  | LEMON | 82.3 ± 0.3| 83.4 ± 0.8| 83.5 ± 0.8| 83.5 ± 0.7| 85.9 ± 2.2|
> | noun  | TRACED - LEMON| 85.2 ± 0.3| 86.1 ± 0.6| 85.0 ± 0.9| 85.7 ± 0.2| 87.1 ± 1.6|
> | noun  | LEMON_OPT| 84.1 ± 0.3| 85.4 ± 1.1| 85.6 ± 0.7| 85.8 ± 0.2| 88.3 ± 1.1|
> | noun  | TRACED - LEMON_OPT| 86.0 ± 0.5| 86.3 ± 0.7| 86.4 ± 0.8| 86.6 ± 0.2| 87.8 ± 1.2|
> | noun  | CLIP  | 83.1 ± 0.2| 83.9 ± 0.7| 84.3 ± 0.4| 84.5 ± 0.7| 86.0 ± 1.2|
> | noun  | TRACED - CLIP | 85.6 ± 0.3| 86.2 ± 1.0| 85.6 ± 0.7| 85.3 ± 0.0| 87.3 ± 2.1|
> | random| BLIP  | 98.3 ± 0.1| 98.7 ± 0.3| 98.4 ± 0.4| 98.5 ± 0.3| 98.5 ± 0.5|
> | random| TRACED - BLIP | 98.8 ± 0.0| 98.8 ± 0.4| 99.1 ± 0.2| 99.0 ± 0.2| 98.5 ± 0.5|
> | random| LEMON | 97.5 ± 0.1| 98.0 ± 0.3| 97.8 ± 0.2| 97.6 ± 0.1| 97.7 ± 0.3|
> | random| TRACED - LEMON| 97.6 ± 0.1| 98.2 ± 0.1| 97.9 ± 0.2| 97.1 ± 0.3| 98.1 ± 0.1|
> | random| LEMON_OPT| 97.6 ± 0.2| 98.1 ± 0.0| 97.7 ± 0.0| 97.5 ± 0.2| 98.0 ± 0.2|
> | random| TRACED - LEMON_OPT| 97.7 ± 0.1| 98.3 ± 0.1| 97.8 ± 0.1| 97.4 ± 0.2| 98.7 ± 0.2|
> | random| CLIP  | 97.5 ± 0.3| 98.1 ± 0.1| 97.4 ± 0.1| 97.2 ± 0.1| 98.0 ± 0.2|
> | random| TRACED - CLIP | 97.7 ± 0.3| 98.2 ± 0.0| 97.6 ± 0.2| 97.3 ± 0.2| 98.0 ± 0.2|

---

### Author Response · Authors · 2025-11-26

We thank the reviewers for their careful and constructive feedback. As pointed out by the reviewers, TRACED is a new trajectory-based approach for error detection. It is model-agnostic, can provide interpretable token-level error information, and shows consistent improvements across multiple baselines.

We have addressed all raised concerns: we added diverse real examples of error types (Reviewer QGRZ), clarified our fine-grained noise construction (Reviewer ZqLU), provided new ablations on the noise levels (Reviewer QGRZ), trajectory length T (Reviewers QGRZ and TbST), candidate count N (Reviewers QGRZ and TbST), semantic similarity choices (Reviewer TbST), included real-data validation through manual inspection (Reviewer TbST), reported downstream captioning performance after filtering and correction (Reviewers ZqLU and PQfZ) and evaluated the interpretable token-level error information of our method (Reviewer PQfZ).

We hope these additions resolve the reviewers’ questions, and we kindly invite reviewers to update their scores if they find our answers satisfactory.

---

### Author Response · Authors · 2025-12-03

The paper has been updated with the new results. We added standard errors over 3 seeds for the downstream captioning experiments (Reviewers ZqLU and PQfZ) as well as for the ablation on the candidate count N (Reviewers QGRZ and TbST).

---

### Meta-Review · Area_Chair_JxKQ · 2026-01-07

**Summary:**

The work proposes an interesting approach for error detection. The idea is quite neat and incorporate look at the trajectory of "correcting" a caption to see if the original caption contained errors or not with the premise that wrong captions require a lot of "corrections". The authors show the advantage of the approach for various noise types that are based on LLM and therefore are more realistic then just random replacements. The work is interesting. I am not sure about the interpretability claim but the work merit publication also if this part of the work is questionable.

**Reviewer Concerns:**

The concerns were about evaluation, understanding better the method and intrepretability. The first two parts were answered well. The last not so much (there is still the question about correlation with human judgment) but the paper can still be accepted.

**Reviewer Scores:**

One of the reviewers explicitly said that the score should be increased so at one reviewer would increase the score. I believe the answers of the reviewers were thorough enough to convince also the other reviewers.

---

### Decision · Program_Chairs · 2026-01-26

Accept (Poster)